# China can be self-sufficient in maize production by 2030 with optimal crop management

Ning Luo [1,2], Qingfeng Meng [1] ✉, Puyu Feng [3], Ziren Qu[1], Yonghong Yu[1], De Li Liu [4,5], Christoph Müller [2] & Pu Wang[1]

Population growth and economic development in China has increased the demand for food and animal feed, raising questions regarding China's future maize production self-sufficiency. Here, we address this challenge by combining data-driven projections with a machine learning method on data from 402 stations, with data from 87 field experiments across China. Current maize yield would be roughly doubled with the implementation of optimal planting density and management. In the 2030 s, we estimate a 52% yield improvement through dense planting and soil improvement under a high-end climate forcing Shared Socio-Economic Pathway (SSP585), compared with a historical climate trend. Based on our results, yield gains from soil improvement outweigh the adverse effects of climate change. This implies that China can be self-sufficient in maize by using current cropping areas. Our results challenge the view of yield stagnation in most global areas and provide an example of how food security can be achieved with optimal crop-soil management under future climate change scenarios.

Maize (*Zea mays* L.) production needs to be substantially increased to address the growing competition for a limited supply of arable land, as well as the increasing demand for food and feed under population growth and economic development[1,2]. A general pattern of declining growth rates in global crop yields is observed in many important producer countries, and raise concerns about whether future output growth can keep pace with demand[3,4]. China is the second largest producer of maize in the world, producing 260 million tons annually (2016–2020) and contributing 23% of the maize supply using 21% of the global maize area[5]. However, maize imports in China saw a rapid increase recently in response to the deficit between growing domestic consumption and stalled production, and peaked in 2021 with 28.4 million tons[6]. Continued high Chinese maize imports may contribute towards increased prices and volatility in global cereal markets in the future. Given the limited possibility for area expansion, the ability of China's maize production to reach self-sufficiency through continued grain yield (i.e., outputs per unit area) improvements therefore plays a critical role for both global food security and ending hunger by 2030–Goal 2 of the 17 United Nations Sustainable Development Goals[7].

Crop yield gains rely on complex interactions between genotypes, environmental factors (including climate and soil conditions), and agricultural management[8]. Recent work in three irrigated maize regions (Lower Niobrara, Tri-Basin, and Upper Big Blue) in Nebraska illustrates that climate trends and agronomic improvements, not genetic improvements, underpin recent maize yield gains[9]. Evidence points towards plant density as one of the critical indicators in explaining maize yield booms in the USA and other parts of the world[10,11]. In North America, optimum plant density (OPD) increased at a rate of 700 plants per hectare per year during 1987–2016[12]. Plant density contribution to maize yield gain ranges from 8.5 to 17%. Brazil's maize productivity has increased due to increased density-tolerance in modern hybrids, with planting density increasing from $7.1 \times 10^4$ plants ha$^{-1}$ in the 1970s to $8.5 \times 10^4$ plants ha$^{-1}$ in the 1990s[13]. Maize yields in France quadrupled in the 1950s–1980s, with yields increasingly

[1]College of Agronomy and Biotechnology, China Agricultural University, 100193 Beijing, China. [2]Potsdam Institute for Climate Impact Research (PIK), Member of the Leibniz Association, 14412 Potsdam, Germany. [3]College of Land Science and Technology, China Agricultural University, 100193 Beijing, China. [4]NSW Department of Primary Industries, Wagga Wagga Agricultural Institute, Wagga Wagga, NSW 2650, Australia. [5]Climate Change Research Centre and ARC Centre of Excellence for Climate Extremes, University of New South Wales, Sydney, NSW 2052, Australia. ✉e-mail: mengqf@cau.edu.cn

correlated with planting density, due to genetic gains[14]. Notably, Chinese farmers have steadily increased maize plant density to pursue greater yields since the 1950s, at an average rate of 500 plants per hectare per decade, despite lower magnitudes of yield and density compared to the USA (Supplementary Fig. 1). Hence, a critical question is whether the strategy that dense planting combined with an improved environment such as soil-management and better cultivars in China is adequate to achieve the anticipated increase in maize yield in the coming decades? The answer to this question is vital for analyzing future domestic and global maize supply-demand balances.

Dense planting would impose competition for water, nutrients, and light[15]. Understanding the OPD that maximizes utilization of available resources for the highest grain yield in each farming system can help uncover potential trade-offs between agronomic inputs and economic gains. Approaches to estimate OPD are limited, especially in terms of coverage of different agro-ecological zones. For example, current on-farm trials that explore yield-density response generally include limited density settings and maize hybrids in specific sites[16], such that results are difficult to extrapolate to all growing environments. Statistical approaches based on large datasets provide more comprehensive insights into yield-density relationships[17], but have considerable uncertainties due to the quantity and quality of the observed data[18]. Machine learning algorithms provide a robust approach to investigate complex interactions between crop management and the growing environment, due to their ability to uncover hierarchical and non-linear relationships between the response variable and predictor variables on the basis of ensemble learning approaches[19,20]. Combining machine learning and testing with sufficient field trials is essential for the accurate prediction of OPD and its associated crop productivity, and thus for the guidance of farmers' practices.

In this study, we first used a data-driven approach based on the Random Forest (RF) algorithm to recognize the precise OPD across China. Building on a database of 2442 paired observations (yield-density points) derived from 125 studies published between 2000 and 2021 across major maize areas in China, a RF algorithm is trained and tested at the nationwide level to predict OPD as a function of environmental inputs, management and soil organic matter content (SOM). The RF predictive model is then run with a large regional-scale dataset (including 402 stations, see "Methods" for details) to assess OPD across China under current conditions. Next, these projected OPDs are compared with results from 87 field trials across China[21]. Finally, projections of maize OPD are performed using a climate change scenario produced by 22 Global Climate Models of the Coupled Model Intercomparison Project Phase 6 (CMIP6) under a high-end radiative forcing setup, the Shared Socio-Economic Pathways (SSP585) in the 2030 s. We explore the potential of yield improvement through dense planting in combination with best-suited hybrid varieties and soil improvements under current and future climates through the integration of machine learning and field trials methodologies. Furthermore, the question of whether China can be self-sufficient in maize under projected future climate change given its existing cropping areas and optimal agricultural management in the 2030 s is also addressed.

## Results

### Data-driven projections and variables

To understand the OPD for obtaining a balance between maize yield and resource cost under different environmental conditions, we develop the OPD-RF model. It includes six indicators, incorporating daily minimum temperature (Tmin), daily maximum temperature (Tmax), precipitation (Prec), solar radiation (Radn), hybrid characteristics such as growing-degree days (GDD) and soil quality indicator (SOM) from a dataset with 448 paired observations in 125 publications conducted over China's major maize cropping areas (Fig. 1a, b; see

"Methods" for details). The OPD-RF model explains 60% of OPD variance across the study region, with a bias relative root mean square error (RRMSE) of 11.9% and root mean square error (RMSE) of $0.9 \times 10^4$ plants ha$^{-1}$ (Fig. 1c), indicating a good performance of the OPD-RF model in modeling OPD.

In addition to evaluating the predictive skill of the OPD-RF model, we also use it to disentangle the relative importance of the drivers that shape OPD patterns in the four regions (Fig. 2). In more than three regions, Tmin, Radn and SOM always rank among the top three among the six explanatory factors. Tmin significantly decreases OPD from 0.16 to $0.51 \times 10^4$ plants ha$^{-1}$ per 1 °C increase while Radn increase 1.00 to $2.00 \times 10^4$ plants ha$^{-1}$ per 1000 MJ m$^{-2}$ increase. In the future, the influence of Tmax on OPD is projected to be strengthened (Supplementary Fig. 2). Notably, higher SOM content over ~20 g kg$^{-1}$ does not continue to enhance OPD (Fig. 2g). We thereby consider improving SOM to the threshold value (i.e., 20 g kg$^{-1}$) as an effective management guideline for density optimization in the subsequent scenario analysis.

The current spatial patterns of OPD simulated by the RF model based on 402 stations located in major maize areas suggests an average of $7.8 \times 10^4$ plants ha$^{-1}$ in density (Fig. 3). Among all regions, Northeast China and North China Plain have a similar OPD to the national average ($7.6 \times 10^4$ plants ha$^{-1}$ for Northeast China and $7.9 \times 10^4$ plants ha$^{-1}$ for North China Plain), while the maximum OPD was $8.6 \times 10^4$ plants ha$^{-1}$ in Northwest China and the minimum was $7.1 \times 10^4$ plants ha$^{-1}$ in Southwest China (Supplementary Table 1). Accordingly, the optimum yield at the OPD is 11.4 Mg ha$^{-1}$ in Northeast China, 11.8 Mg ha$^{-1}$ in North China Plain, 12.6 Mg ha$^{-1}$ in Northwest China and 10.9 Mg ha$^{-1}$ in Southwest China (Fig. 3). There are considerable gaps between farmers' currently planted densities and OPD over the study regions, whereas farmers achieve 77% of OPD with a range from $1.7 \times 10^4$ plants ha$^{-1}$ for the North China Plain to $2.3 \times 10^4$ plants ha$^{-1}$ for the Southwest China. Furthermore, closing the density gap could improve national maize yield by an average of 95%, compared to current levels (6.0 Mg ha$^{-1}$)[22].

### Field observations for grain yield with OPD

To verify the feasibility of OPD in field, we investigate OPD and yield through 87 field trials (site × years) across the major maize-producing area in China during 2017–2020 (Fig. 4). Two treatments in each site are designed and compared (Supplementary Data): (i) the control: using the local maize hybrid and farmers' planting densities, and (ii) optimum treatment: the OPD is defined as the average density for a group of five high-yielding hybrids at the highest grain yield from the density experiment.

Results from the optimum treatment in field experiments at the national scale are consistent with RF projections with an average plant density of $7.8 \times 10^4$ plants ha$^{-1}$ and grain yield of 11.7 Mg ha$^{-1}$ (Supplementary Table 2). North China Plain has the highest OPD ($8.3 \times 10^4$ plants ha$^{-1}$) and Southwest China saw the lowest ($6.6 \times 10^4$ plants ha$^{-1}$). Northeast China experiences the highest grain yield among the three regions (12.5 Mg ha$^{-1}$), because the spring-maize cropping system and long-maturing hybrids are popular in this region. Compared to the control treatment, maize yield could be intensively improved by an average of 21% by adopting the optimum density together with appropriate hybrids, nearly doubling the current farmers' yield[22], and staying consistent with the RF predictions. Furthermore, the yield gain from genetic and density improvements is 5.9% and 7.3%, respectively (Supplementary Fig. 3). The density × genetics interaction contributes 7.4% of yield improvement.

### Grain yield under future climate and soil improvement

We next derive the plant density response on the basis of the trained RF model under future climate conditions following a high radiative forcing scenario (SSP585) and a scenario focusing on SOM improvement. At current soil conditions and crop management strategies,

OPDs tend to decline across all regions under climate change in the next 20 years, with larger rates concentrated in the Northeast China (−2.2%) and North China Plain (−3.5%) (Fig. 5 and Supplementary Table 3). By the 2030 s, average OPDs in China decrease by 1.6% due to climate change. To determine the implications of soil improvement in OPDs and grain yield, we apply the RF model with soil improvement (i.e., optimizing SOM to 20 g kg$^{-1}$) under current and future climates. On average, a 2.5% increase in OPDs from SOM optimization compared to simulation at current soil conditions is observed.

The rapid growth of maize imports suggests that the capacity of maize production in China is struggling to meet domestic population demands (Supplementary Fig. 4). Based on historical trends (Supplementary Fig. 1), we estimate constant increases in plant density and yield from the 2010s to 2030 s, resulting in a national plant density of 7.1 × 10$^4$ plants ha$^{-1}$ and yield of 7.7 Mg ha$^{-1}$ in the 2030 s (Fig. 5c, d). Projections using the OPD-RF model with soil improvement in climate change indicate that grain yield would be increased by 52% (average 11.7 Mg ha$^{-1}$) in the 2030 s, compared to yields at the historical trend. Assuming current trends in the harvested area remain unchanged, our OPDs scenario results in 492 Mt of maize production by 2035, which

would be able to meet 100% of national demands (292 Mt, see "Methods").

## Discussion

As the second largest producer in the world, maize production is essential in China for the global maize supply-demand balance. Planting density can mediate between genotype, environment and management. Planting at OPD is a critical management decision that has contributed to continuous maize gains in the past[23]. Thus, understanding and quantifying OPDs across major maize areas in China is essential to yield improvement and national food security. Our study shows that OPDs are much greater than current farmer practices under current and projected climates. The OPDs under the current climate ranged from 7.1 × 10$^4$ plants ha$^{-1}$ in Southwest China to 8.6 × 10$^4$ plants ha$^{-1}$ in Northwest China across the study regions with an average of 7.8 × 10$^4$ plants ha$^{-1}$ (Fig. 3), comparable to findings in other maize-producing regions. In Brazil, the average optimum plant density for maize hybrids released during the 1970s–1990s is 7.8 × 10$^4$ plants ha$^{-1}$, ranging from 7.1 to 8.5 × 10$^4$ plants ha$^{-1}$ [13]. Similar evidence is also observed in the European Union (EU), which has an intensive maize

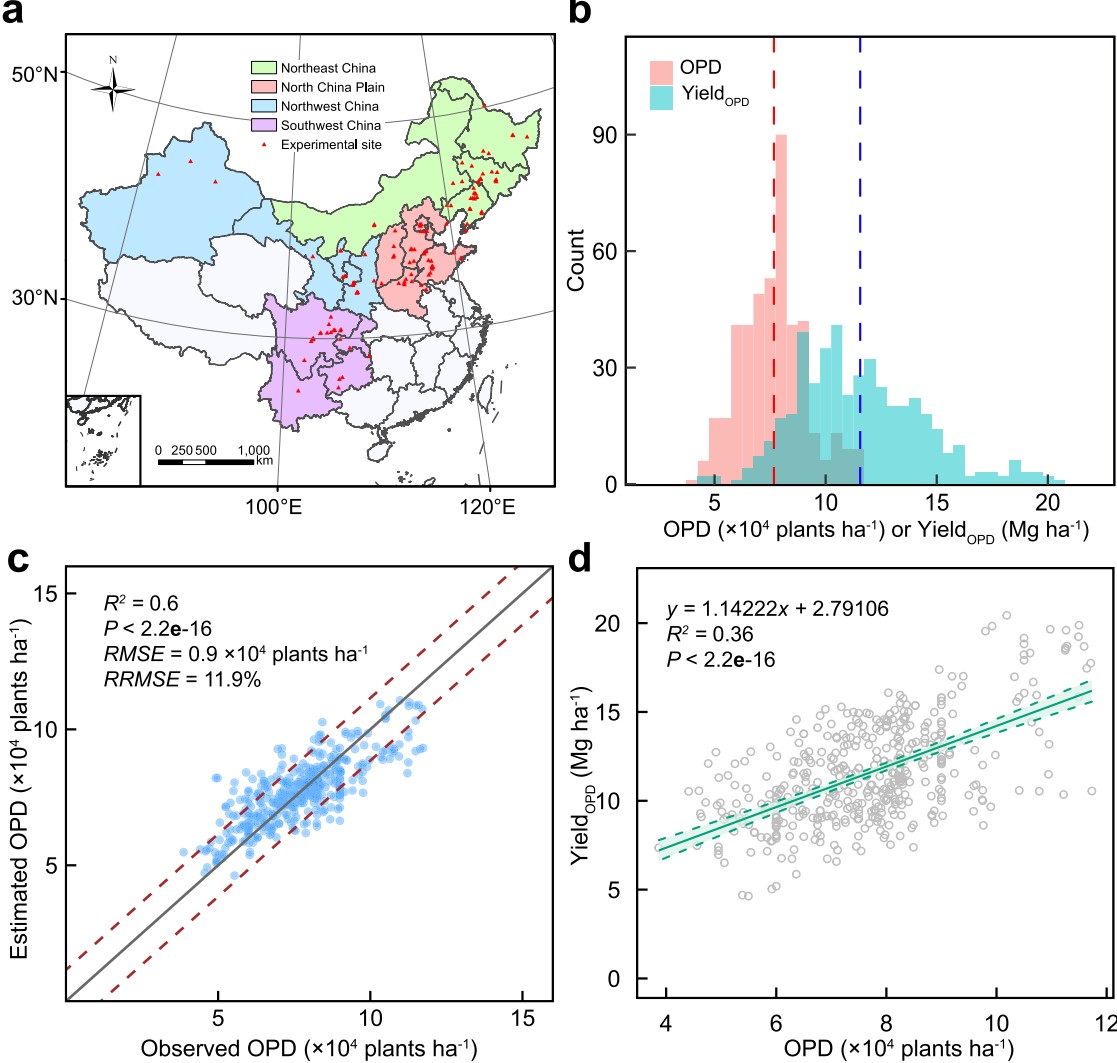

**Fig. 1 | The random forest (RF) model development. a** Study area and locations of experimental sites found in the literature. **b** Data distributions of OPDs and Yield$_{OPD}$ estimated from literature data with yield-density quadratic model. **c** Comparisons of the estimated OPD by RF model with observed OPD in 448 observations. The dashed line represents the 15% error line. **d** Linear-model for the relationship between OPD and Yield$_{OPD}$. We find a similar performance in linear and quadratic fits for the relationship between the Yield$_{OPD}$ and OPD and the linear nature is applied. The solid line is the regression lines and the dashed line indicate the 95% confidence interval. Statistical significance in **c** and **d** is obtained with a two-tailed Student's $t$ test. Source data are provided as a Source Data file.

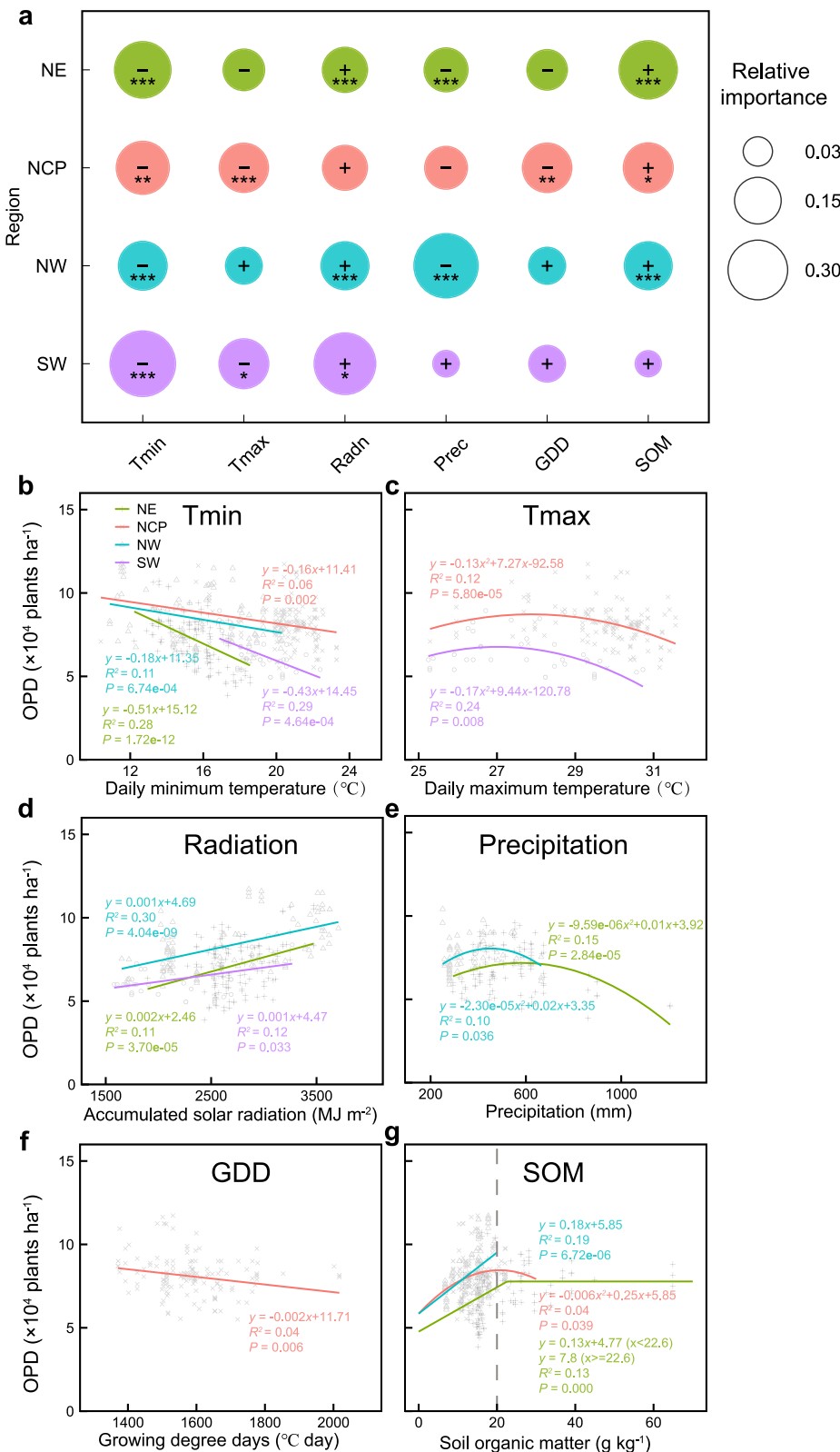

**Fig. 2 | Importance of variables influencing OPD and relationships between drivers and OPD for each region based on observations from the literatures in 2000-2021. a** The relative importance of variables influencing OPD in this study. The rows show the results for each region (NE: Northeast China, NCP: North China Plain, NW: Northwest China, SW: Southwest China). The circle size should be compared only within a row. The importance of each variable is expressed as the mean increase in prediction error (that is, the increase in mean square error, % IncMSE) with predictor omitted, scaled to sum to 100% for each analysis. The symbols + and − indicate positive and negative effects of the variables on OPD, respectively. The asterisks indicate the statistical significance of the effect (*$P$ < 0.05; ** $P$ < 0.01; ***$P$ < 0.001). The best fitted functions for relationship between OPD and each driver are shown in **b**–**g** for all significant relationships. Panels represent Tmin (**b**), Tmax (**c**), Radiation (**d**), Precipitation (**e**), GDD (**f**), and SOM (**g**). Statistical significance is obtained with a two-tailed Student's $t$ test. Source data are provided as a Source Data file.

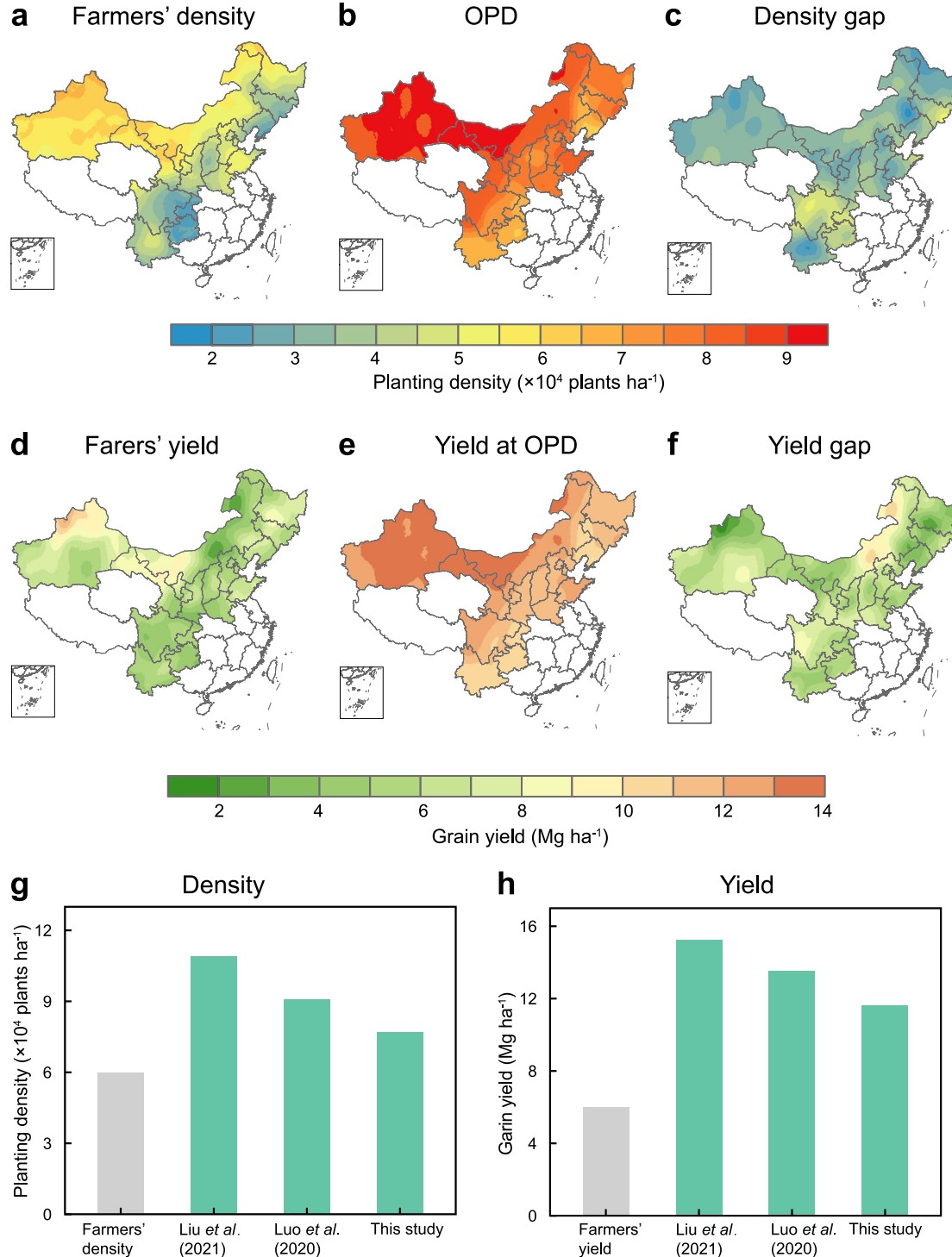

**Fig. 3 | Optimum density and yield from RF model and in comparison with farmers' practices and two other methods.** Spatial distribution of famers' planting density (**a**), OPD (**b**), density gap (**c**), current yields (**d**), yield at OPD (**e**), and yield gap (**f**). Farmers' planting density is derived from Ming et al. Current yield is collected from the public dataset[22]. Density gap is equaled to OPD minus farmers' densities. Yield gap is defined in an analogous manner. Comparison of maize density (**g**) and yield (**h**) under different methods. Source data are provided as a Source Data file.

cultivation system with plant densities up to $8.0 \times 10^4$ plants ha$^{-1}$ in fertile growing areas[24]. The average OPD from our RF model is lower than that of the North America region, where simulated OPD is at an average of $9.3 \times 10^4$ plants ha$^{-1}$ for 2012–2016[12].

Recently, machine learning techniques have been successfully implemented in agricultural production to investigate various agronomic indicators (e.g., crop yield)[20,25,26]. As a popular decision-tree-based ensemble machine learning algorithm, RF can handle nonlinear effects and complex interactions among variables[27]. Through the implementation as an RF model, the OPD projection is data-driven and does not rely on pre-specified equations or functional form. Here, we combine large, station-scale datasets on weather, management and soil condition, and the RF algorithm to identify OPDs over China's maize areas. Previous works based on statistical models often

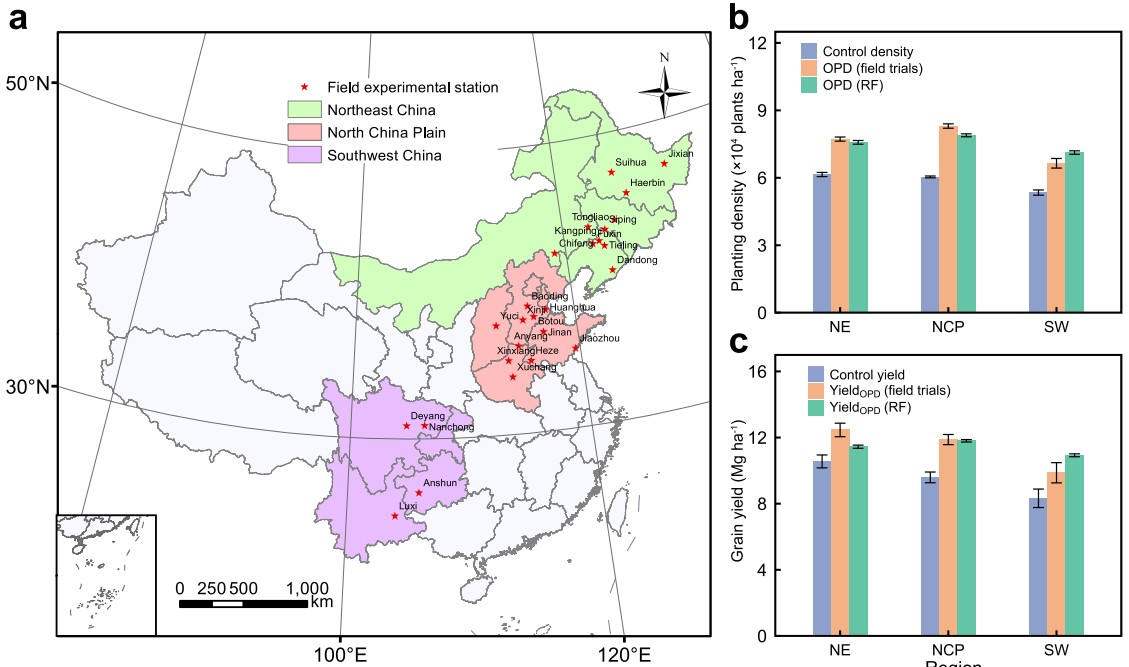

**Fig. 4 | Optimized density and yield in field experiments. a** Distribution of experimental stations for field trials across three regions. **b**, **c** Comparison of maize optimal plant density (OPD) and grain yield (Yield$_{OPD}$) between field trials and OPD-RF simulation (RF). Data are presented as mean values ± standard error. Control density: density at the control, OPD (field trials): optimal plant density from optimum treat in field trials, OPD (RF): optimal plant density from OPD-RF simulation. Control yield: grain yield with density at the control, Yield$_{OPD}$ (field trails): grain yield with density at optimum treat, Yield$_{OPD}$ (RF): grain yield with OPD from RF simulation. NE Northeast China, NCP North China Plain, SW Southwest China. Source data are provided as a Source Data file.

overestimate the OPD in the absence of genotype (G) × environment (E) × management (M) interactions (Fig. 3g, h). Liu et al. shows that OPD simulated from the solar radiation-based linear model average 10.9 × 10⁴ plants ha⁻¹ across China[28]. Likewise, a recent study synthesizing 157 scientific publications conclude that the OPD generally averaged 9.4 × 10⁴ plants ha⁻¹, relying on quadratic-curves models[29]. The methods applied in these previous studies are relatively easy to handle and compute. However, they have limitations on specific-OPD predictions due to potential over-simplification in describing interactions between climatic and agronomic conditions[10]. Several aspects reinforce the reliability of our findings drawn from the framework. Firstly, we calculate OPDs from various field trials following quadratic models in the training dataset[17]. Secondly, the RF model contains the multi-density related indicators and is in line with current knowledge on maize physiology that optimal planting density varies relative to weather, soil condition and crop breeding[13,30]. With a *RRMSE* of 11.9% and $R^2$ of 0.6 (Fig. 1c), the good performance of our model suggests that the combination of climate-soil datasets within the maize growing season with machine-learning technique is a promising method to investigate the impact of climate on OPDs.

Indeed, negative impacts of climate change are predicted for most sites across China, though the magnitudes of OPDs and grain yield changes vary substantially across regions (Fig. 5a, b). This finding suggests that achieving high yields with dense planting would become more challenging under future climate change, especially in the North China Plain. Improving soil quality would alleviate climate impacts, by shifting maize yield-density relationships[30]. Based on knowledge of the relationship between OPDs and SOM (Fig. 2g), we estimate that declines in OPDs from climate impacts could be offset by soil improvement (i.e., via SOM). On the other hand, dense planting influences plant architecture, growth and developmental patterns, encouraging maize breeders to produce new genotypes with ideotype[23,31]. For example, compared to older maize hybrids, modern maize hybrids in Brazil have compact canopy architecture with shorter plants, fewer and more up-right leaves and enhanced light interception, thus achieving high yield at higher plant densities[13]. Further studies are necessary to better understand the contribution of interactions between genotype, environment and management to changes in OPD.

Although our results are theoretically feasible for farmers adoption and confirmed by multi-field trials (Fig. 4), several factors limit farmers' planting density decisions (Supplementary Table 4)[32,33]. Increasing temperature will increase atmospheric water demand which could lead to drought stress from increased vapor pressure deficit (VPD), subsequently reducing plant density and decreasing yield across most maize areas[34]. Low temperatures at the earlier growth stage together with water stress in the Northeast China is a critical limitation for maize density improvement[35]. Solar radiation, which influences photosynthesis in plant leaves as the energy source in crop production[36], has decreased over the past decades in the North China Plain[37], potentially limiting further increases in plant density and maize yield. A lack of irrigation in the Northwest China and diverse landforms and ecosystems in Southwest China were the major restrictions to increasing plant density in these regions[38,39]. In addition, increased risks of lodging arising from greater plant densities influence farmers' sowing density selections[40]. Co-efforts from breeders and producers will determine further increases in China's maize planting density and yield. An ideal plant architecture with an appropriate canopy structure that intercepts more solar radiation is crucial to dense planting and achieving high yields[31]. Tian et al.[41] report that upright plant architecture in modern hybrids provides opportunities for dense planting, providing new insights into high-density-yield maize breeding. As a follow-up, practices must be evaluated and modified to match high-yielding systems with optimum planting density.

The achievement of modeled OPDs would need to be supported by enhanced management strategies (e.g., irrigation and fertilization). Increasing planting density increases plant-to-plant competition in high-yielding systems[42] and induces greater sensitivity to drought in

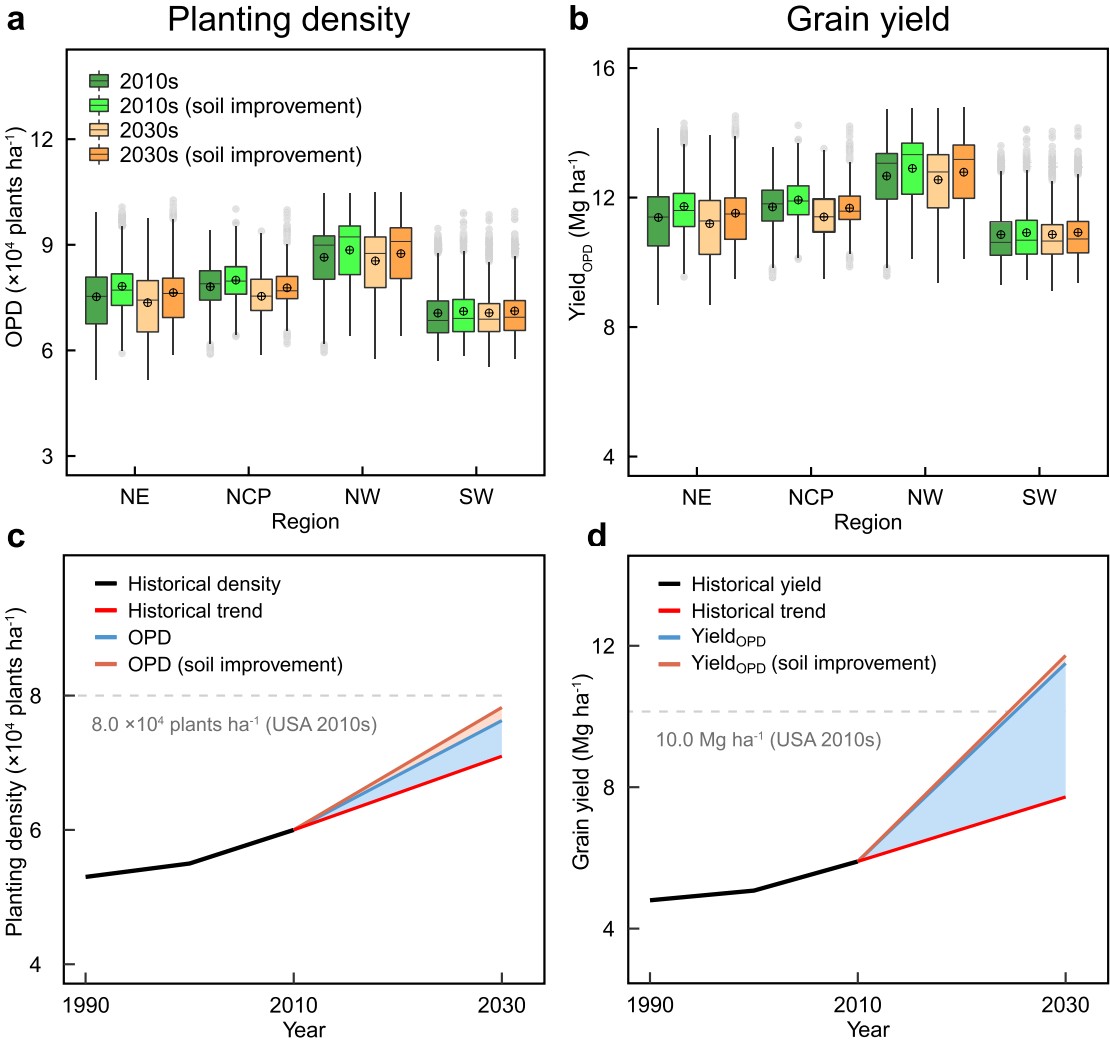

**Fig. 5 | Maize density and yield projections based on different scenarios.** Upper panels show average density (**a**) and yield (**b**) in the 2010s (2010–2019) and 2030 s (2030–2039) based on climate and climate coupled to SOM improvement projections. Box boundaries indicate the 25th and 75th percentiles across 22 GCMs, and whiskers below and above the box indicate the 10th and 90th percentiles, respectively. The black lines and crosshairs within each box indicate the median and mean values, respectively. Lower panels show density (**c**) and yield (**d**) for China by 2030 s compared with current conditions in the US. The black line represents average data from framers' practice. The red, blue, and brown line represent density (yield) predicted under historical trend, optimum treatment, and optimum treatment coupled to SOM improvement, respectively. NE Northeast China, NCP North China Plain, NW Northwest China, SW Southwest China. Source data are provided as a Source Data file.

maize[34]. Drought is one of most important limitations to yield gains in 40% of maize area in China[21] and irrigation should be considered in the intensification of maize production systems, which has been shown to be an effective management practice for improving yields[43]. Meanwhile, nitrogen fertilizer applied at the critical stages for high-planting density is required for optimal maize yield[44]. To this end, an integrated crop-soil system approach in management will be required in the future to support high-yielding maize system with dense planting[45].

Uncertainties exist in estimating OPDs and grain yield in our work. Although the OPDs predictive model is developed with an RF algorithm based on data collected from 125 published studies containing multi-density settings, we recognize that evaluation of OPDs can be improved by more models that couple biophysical modeling and machine learning[46]. In addition, our simulation presents some limitations by neglecting several factors, particularly water stress and nutrient deficit due to increasing densities[23]. Here, we focus on studies conducted under no water nor nutrient deficits and the premise that all sites would take measures to maintain adequate water and fertilization availability, simplifying real-world density responses. This implies that

efforts to increase planting densities are not taken in isolation but combined with other management aspects, such as adequate fertilization, to expand the potential of planting higher densities. Continuous efforts are needed to improve the OPDs prediction by considering other possible factors that this study cannot address, such as risks of lodging, pest and diseases. Despite these limitations, we find great potential to meet the required yield increases that would allow for self-sufficient maize production in 2035 by optimizing plant density under future climate change.

Overall, maize yield improvement depends on complex interactions among genetics, environment and management. Based on the data-driven approach and field trials, we demonstrate that China would be self-sufficient in maize with current cropping areas in the 2030 s through denser planting in combination with selecting best-suited hybrid varieties and soil improvements under future climate. The findings also indicate that high-quality soils with higher SOM could moderate the impact of climate change on OPD and thus improve maize yield. The OPD reflect the density-genetics interaction in this study. In the future, the interactions between hybrid and density could

be further enhanced for yield. Our results challenge the view that grain yield have reached an attainable maximum in most global areas and provide a workable example for grain yield improvement through dense planting.

## Methods

### Study area

China is the second major producer of maize, constituting one-fifth of global production (Supplementary Fig. 5). Generally, maize agro-ecological regions in China could be divided into four regions based on climate, management and maize cropping pattern from north to south[35]: Northeast China (NE), North China Plain (NCP), Northwest China (NW), and Southwest China (SW). The study area in Fig. 1a covered 91% of the national maize cropping areas and produced roughly 90% of national output[22]. Because of the diversity of climate and management among these four regions, the study area serves as an excellent laboratory for further improving maize yields.

### Data collection for RF training

The literature database was derived from 125 studies conducted in the study area published. The literature search was performed using the China National Knowledge Infrastructure and the Web of Science for relevant papers published between January 2000 and October 2021 using the following keywords: "density*" AND "yield*" AND ("maize*" OR "corn*") AND ("China*" OR "Chinese*"). In total, 257 scientific articles are retrieved, and review of each study is carried out based on (i) experiments conducted in the field; (ii) at a given field trial, more than three levels of plant density evaluated; (iii) soil properties and management information, in particular, water condition, planting and harvest date reported. Most of the data are retrieved from tables directly. In case some data were only presented in figures, values are extracted using the WebPlotDigitizer software[47]. After the review, a total of 125 published studies with 151 site-years and 2442 paired observations are considered eligible (Supplementary Fig. 6 and Supplementary Date). Given that soil organic matter (SOM) is a critical indicator of soil properties associated with higher fertility[48], we here take it as a representation of soil condition in our further analysis. Data on SOM in the 0–20 cm soil layer, yield at each density, descriptions of treatments (e.g., hybrids, sowing and maturity date), and locations of experimental sites were collected. Data on climatic factors including daily temperature, total precipitation, and solar radiation during the maize growing season were also extracted.

Growing degree days (GDDs) are generally taken as a measure of the thermal time required for a specific cultivar to develop[49]. The daily heat unit, $GDD_d$, is calculated according to daily maximum ($T_{max}$) and minimum temperature ($T_{min}$) and defined as follows (Eqs. (1) and (2)):

$$GDD_d = \frac{T^*_{min,d} + T^*_{max,d}}{2} - T_{low} \qquad (1)$$

where,

$$T^*_{max,d} = \begin{cases} T_{max,d}, & if\ T_{low} < T_{max,d} < T_{high}, \\ T_{low}, & if\ T_{max,d} \le T_{low}, \\ T_{high}, & if\ T_{max,d} \ge T_{high} \end{cases} \qquad (2)$$

$T^*_{min,d}$ is defined by using the same low and high bounds of $T_{low} = 10\ °C$ and $T_{high} = 30\ °C$.

OPD is a critical management decision for crop yield[15]. Previous research has shown that the quadratic model performs better in depicting maize yield responses to plant density[17,29]. Here, we use quadratic curves to estimate OPD and corresponding yield at each OPD (Yield$_{OPD}$) for each specific trial. In total, 448 (site × year × hybrid) paired observations are collected, providing the basis for the development of the Random Forest (RF) model with OPD (Fig. 6).

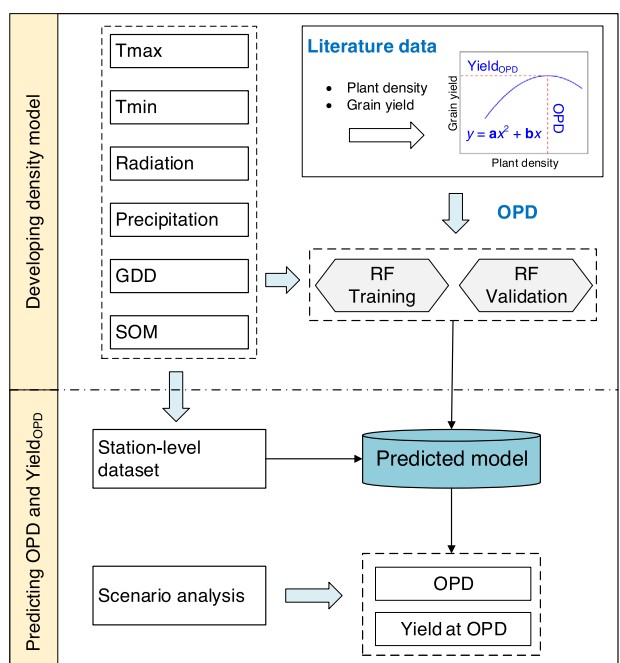

**Fig. 6 | Framework of the procedure for mapping regional optimum plant density (OPD) and yield at OPD (Yield$_{OPD}$).** Tmax: daily maximum temperature. Tmin: daily minimum temperature. GDD: growing degree days (10–30 °C). SOM: soil organic matter. Tmax and Tmin is averaged during maize growing season, and solar radiation, precipitation, and GDD is the sum of growing season.

### RF modeling framework

As a nonparametric and ensemble learning algorithm originating from classification and regression trees, RF is now increasingly used in the crop sciences[50]. Given the better performance of the RF model on agricultural-based applications, along with its ability to provide the relative importance of each predictor in determining response variable[20,26,46], we hereby take advantage of the RF model to assess maize optimum plant density at each site. The RF model is performed in R software using the "*RandomForest*" package with default settings, *mtry* (the number of randomly selected predictor variables at each node) = 3 and *ntree* (the number of trees to grow in the forest) = 500. The relative importance of variables is estimated using the "*%IncMSE*" metric in the RF model.

We use four agro-climatic indicators (i.e., daily minimum temperature, daily maximum temperature, precipitation and radiation), GDDs and SOM from the literature database (448 paired observations) to train the RF model (Fig. 6). For RF model calibration and validation, 80% of each dataset is randomly selected for model training and the rest 20% of the dataset is used for model performance evaluation. 'Leave-one-out' prediction is conducted to critically evaluate the reliability of the surrogate models. The coefficient of determination ($R^2$, Eq. (3)), mean root mean square error (RMSE, Eq. (4)), and relative Root Mean Square Error (RRMSE, Eq. (5)) are used to evaluate model performance. These indices are calculated as follows:

$$R^2 = \left( \frac{\sum_{i=1}^{n}(O_i - \bar{O})(S_i - \bar{S})}{\sqrt{\sum_{i=1}^{n}(O_i - \bar{O})^2}\sqrt{\sum_{i=1}^{n}(S_i - \bar{O})^2}} \right)^2 \qquad (3)$$

$$RMSE = \sqrt{\frac{1}{n}\sum_{i=1}^{n}(S_i - O_i)^2} \qquad (4)$$

$$RRMSE = \frac{\sqrt{\frac{1}{n}\sum_{i=1}^{n}(S_i - O_i)^2}}{O_{mean}} \times 100\% \tag{5}$$

where $O_i$ and $S_i$ are the observed and simulated values, respectively; $O_{mean}$ is the average values, respectively. $n$ is the number of samples. $\bar{O}$ and $\bar{S}$ represent the means for the observed and simulated OPD.

## OPD projection under current and future climate data
Data on farmers' planting densities and yields are included to evaluate the yield gain with dense planting. Here, we establish a dataset collected from 402 stations with available weather data, farmers' yield and planting density. Farmers' planting density (actual density) for each site is collected through surveys of the constant 24 farmers in different experimental sites in 2009–2016[32]. In each site, the actual yields are calculated as the average yield over the ten years from 2010 to 2019 obtained from the China Municipal Statistical Yearbook of the National Bureau of Statistics[22]. As the input indicators in the RF-OPD model (Fig. 6), climate data, maize phenology, and soil data are essential. Daily climate data (maximum temperature, minimum temperature, precipitation, sunshine hour) for the period 2000–2020 are directly collected from the China Meteorological Administration (CMA). Daily solar radiation is calculated from observed sunshine hours using the Angstrom-Prescott equation[51]. Data on soil organic carbon (SOC) for each station are collected according to National Earth System Science Data Center[52]. The SOC is converted to SOM by multiplying the factor of 1.724 (SOC% × 1.724)[53]. Observed data on maize growth (including sowing and maturity dates) from 2010 to 2018 are collected from the national agro-meteorological experiment stations across the maize cultivation areas in China[54].

To predict OPD under climate warming, future scenario data is obtained from Global Climate Models (GCMs), which was contributed by the World Climate Research Program (WCRP) of Coupled Model Inter-comparison Project Phase 6 (CMIP6, https://esgf-node.llnl.gov/search/cmip6). Daily weather data (daily temperature, precipitation, and solar radiation) from 2010 to 2039 for 22 GCMs (Supplementary Table 5) under a high radiative forcing Shared Socio-Economic Pathway (SSP585) was downscaled from monthly gridded data using the statistical downscaling model NWAI-WG[55]. We simulated OPD and Yield$_{OPD}$ for the current period 2010–2019 (abbreviated as 2010s) and future period 2030–2039 (abbreviated as 2030 s) with the framework in Fig. 6.

## Field trials for plant density optimization
In total, 87 field trials were conducted with various plant density settings from 2017 to 2020 in the Chinese Maize Belt (97.5°–135.1° E, 21.1°–53.6° N). This region generally serves as a superb laboratory for exploring maize yield-density relationships in China, due to its large extent from southern tropical and sub-tropical systems (SW) at low latitudes to cool-temperate systems (NE) at high latitudes[56].

In this study, two treatments are designed and compared: (i) the control (CK), which used the local hybrid planted at farmers' cropping density, and (ii) optimum treatment (OT), where the optimal density is defined as the average of plant densities under maximum yield from a group of five high-yielding hybrids (Supplementary Data). At each site, field trials were conducted with three replicates and designed with a split-plot. Maize yield was measured in each plot at physiological maturity and grain was dried in an oven (14% grain moisture).

## Maize demand by 2035 in China
In our study, current (2021–2022) annual domestic maize demand is set as a baseline. We estimated current national maize demand as the average annual national maize production, imports, exports and stock change (i.e., "ending stocks" minus "beginning stocks") during 2021–2022[57]. Future maize demand was projected

for the year 2035, by multiplying the projected population derived from the medium fertility variant[58] by the per capita maize demand, assuming that per capita maize demand is constant at the current (2021–2022) level. China's population is predicted to increase from 1.428 billion in 2021 to 1.434 billion in 2035[58], resulting in 292 Mt of maize demand by 2035.

## Reporting summary
Further information on research design is available in the Nature Portfolio Reporting Summary linked to this article.

## Data availability
The data supporting the findings of this study are available within the paper and its Supplementary Information and source data files. The literature search is performed using the China National Knowledge Infrastructure (https://www.cnki.net) and the Web of Science (http://www.webofknowledge.com). Climate, soil and maize yield are publicly available from the following sources: historically daily weather data directly collected from the China Meteorological Administration (http://www.nmic.cn); the future scenario climate data are at https://esgf-node.llnl.gov/search/cmip6; soil data are available at http://soil.geodata.cn/data/; and maize yield are at https://data.stats.gov.cn. Data for OPD-RF model training and prediction are available on Zenodo repository: https://doi.org/10.5281/zenodo.7857034. Source data are provided with this paper.

## Code availability
R version 4.2.0 is used to train model, aggregate and analyze all results, with packages including 'randomForest (4.7-1.1)', 'tidyverse (1.3.2)', and 'Hmisc (4.7-1)'. The detailed R code for data processing and figure creation are available at Zenodo repository: https://doi.org/10.5281/zenodo.7857034.

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

## Acknowledgements

Acknowledgement for the data support from "Soil SubCenter, National Earth System Science Data Center, National Science & Technology Infrastructure of China. (http://soil.geodata.cn)". We acknowledge all those who participated in the local farmer survey and field trials. We thank David Meng-Chuen Chen for editing to improve the earlier version of the manuscript. We also extend our gratitude to Yupeng Zhu for generously providing the photo of plant morphology observed during the field trials. This work is financially supported by the Key Research and Development Program Project in Hebei Province (22326408D, Q.F.M.), the National Key Research and Development Program of China (2016YFD0300300, P.W.), and the 2115 Talent Development Program of China Agricultural University (Q.F.M.).

## Author contributions

Q.M. conceived and designed the research; N.L. performed the analysis; N.L., Z.Q., and Y.Y. collected the data; P.F. and D.L. provided climate model projections; P.W. provided some discussions; C.M. provided comments on the original draft; N.L. and Q.M wrote the paper with contributions from all authors.

## Competing interests

The authors declare no competing interests.
