## [Peer Review File · Nature Communications]

Reviewers' Comments:

Reviewer #1:

Remarks to the Author:

Thank you for giving me the opportunity to review the article "Title: Can Chinese maize be self-sufficient in climate change with the 2 existing are in the 2030s? Evidences from data-driven projection 3 and field trials" in Nature communication journal.

Reviewer 1

In this manuscript, the authors aim to analyze the random forest machine-learning model to identify optimum plant density (OPDs) over China's maize areas using four different regions. The overall intention of this submission is a good one.

1. Interested readers are expected to quickly catch your contribution, it would be better to highlight major difficulties and challenges, and your original achievements to overcome them, in a clearer way in abstract, introduction and conclusion. Explain key findings/ results in the conclusion section. The English in the present manuscript is not of publication quality and requires major improvement. English editing is required for the whole manuscript content.

Minor changes and editing is required for the following sentences:

Line no-77. Hence, a critical question is whether the strategy that dense planting combined with an improved environment.

Line no-99, the RF algorithm was (firstly) trained and tested at the nationwide level to predict

Line no-216 condition (s)

Line no-233 the global maize supply-demand balance. As an important indicator, plating (?

Planting or what) density

Line no-331 As a follow-up, the practices must be evaluated and modified matching high-yielding systems with optimum planting density.

Line no-338 originated from classification and regression trees that is now increasingly used in ... ?

Line no- 430 Maize demand for by 2035 in China

Line no-394 I strongly recommended reframing the sentence above. for e.g.: Here, 402 stations where weather data and farmers' yield are available were selected to establish a regional-scale dataset.

Need to be short few sentences: (Very long sentences).

Fig. 4 | Optimum density and yield from RF model and its comparison with farmers' practice and other two methods. Spatial distribution of famers' plant density (a), OPD (b), density gap (c), farmers' yield (d), yield at OPD (e), and yield gap (f). Farmers' density was derived from Ming et al. (2017). Farmers' yield was collected from the public dataset²². Density gap equals to OPD minus farmers' density. Yield gap is defined in an analogous manner. Comparison of maize density (g) and yield (h) under different methods.

- What is Comparison in Fig 4.? IS it Comparison?
- What is the main reason for selecting four regions (Northeast China, 330 NE; North China Plain, NCP; Northwest China, NW; Southwest China, SW)? What are the advantages of adopting these locations over others in this study? How will this affect the results? More details should be furnished.

Modeling flowcharts for the developed methodology shall be carefully added to the revised manuscript.

Make sure that all references within the style of the journal.

Reviewer #2:

The manuscript entitled "Can Chinese maize be self-sufficient in climate change with the existing area in the 2030s? Evidences from data-driven projection and field trials" addressed a very important issue on the global largest crop in total production such as maize production under climate change. I was very impressed with this innovative study because of the similar results from two independent methodologies. While a lot of current researches have been applying the machine learning method for crop production prediction, there are rarely field-based observations, especially across wide ecological regions. This study will fill this research gap and supply an excellent example for the combination of two methodologies. The results showed a 55% yield improvement with dense planting in climate change in 2030s in climate change, and the author found the self-sufficient supply for Chinese maize with existing arable land. Interestingly, yield gains from soil improvement with high SOC was more than the negative impact by climate change in next decades. Overall, the result of the manuscript is an important scientific advancement and finding. Before the manuscript could be accepted for publication, it need a major revision for further improvement, especially in the methods section for readers to early follow and understand.

The following comments and suggestions are provided for your consideration.

1. Lines 27-29. "we estimated a 55% yield improvement through dense planting together with soil improvement compared with the historical trend in the climate under high Shared Socio-Economic Pathways", this sentence should be revised. After reading the whole manuscript, I finally understand this sentence. Please revise, especially with the followed sentence "Maize yield was estimated to roughly doubled than current farmers with rational dense planting". It would be better to distinguish the two yield improvements in different period.
2. At the end of the introduction, the objective of the study should be described further. This will help the readers to understand the whole story of the manuscript.

3. Line 53-54 “The gain of crop yield relies on complex interactions among genotype, environment (including climate and soil conditions), and management.” Please add the reference for this statement.
4. Line 66 Fig.1 is not the main results. It’s just a background for maize production between China and US. I suggest to remove Fig.1 to the supplementary information.
5. Line 101-102 This looks like the discussion of your study, I think the author should emphasize your research objective in this section as I mentioned above.
6. Line 247 What reasons may cause the average OPD from our RF model is lower than that of the North America region.
7. Line 249-L253 Machine learning techniques have been successfully implemented in agricultural production to investigate various agronomic indicators (e.g., crop yield), Why do you choose the machine learning techniques? What are its advantages? It would be good to add some discussion here.
8. Line 328-331 It should be stated here how this database was built, instead of stating China is the second major producer of maize, constituting one-fifth of global production, the study region had better use a separate paragraph description.
9. Line 335-336 Case writing should be standardized.
10. Line 363-364 “we found a linear relationship between YieldOPD and OPD across all regions (Fig. 2d).” This is the result of the article and should not appear in materials and methods.
11. Line 393-415 should be improved and expressed clearly, for example, the prediction was station-level, however, the Supplementary Figure 3 was county-scale, which is confused.
12. Lines 397-399. “The average yield was calculated as the average yield over the ten years from 2010 to2019 obtained from the China Municipal Statistical Yearbook of the National

Bureau of Statistics⁴¹ to represent actual farmers' yield in each site", why not directly use the yield of 402 stations, but use the yield in the statistical yearbook.

13. Line 429 "Maize demand for by 2035 in China", should be "Maize demand by 2035 in China"

14. Line 436-437 please show the data source for the population. If possible, please add a reference.

Reviewer #3:

Remarks to the Author:

The authors estimated a 55% maize yield increase in China by 2035 if two conditions are met: increased plant density and soil organic matter improvements. This estimate was derived from a machine learning model, previously calibrated and tested for China.

While I do agree the yields will continue to increase by 2035, I am skeptical about the 55% estimate for three reasons: 1) plant density changes occur concurrently with changes in maize hybrids, and you can't separate density from genetics with this dataset. You modeled the density-genetics interaction, not just the plant density. The citation "8" refers to a well-irrigated region (Nebraska), not reflecting the entire Corn Belt; 2) to realize future yield improvements due to crop improvement (density x genetics changes) other management aspects should be altered in the production system to support the new potential (irrigation, fertilization). This is somewhat discussed in the paper, but it is not highlighted enough; 3) how realistic is it to expect soil improvements within a 10-yr period? Besides biochar application, I don't see any other way to increase soil organic matter within such a short time period.

Line 59 – related to my earlier concern, the authors cite "11" which refers to optimum plant density increase but failed to report a key message from that paper that "plant density contribution to maize yield gain ranged from 8.5% to 17%".

Line 72 and many other places – explain acronyms in the first mention.

Fig 1. Interesting presentation. FYI, according to USDA-NASS statistics, plant densities have been stabilized (or the increase has slowed down) over the last 5 years in the US Corn Belt, yet the grain yield continues to increase.

Line 113. Is GDD for the growing season? If yes, I assume you have planting date data and if that is the case, I wonder about the role of planting date on the optimum plant density.

Fig 2d, the linear nature of the relationship is interesting. Would be good to expand/explain this.

Line 133 change improvement to increase.

Fig 3. Please specify years (time period) of observations, I assume this is for the past years. Also, update the graph by adding the future weather years, probably in the supplemental materials?

Why the current plant density is low in China? Must be a reason. Water limitation?

Fig 6 great presentation.

Line 287 – 289 can't agree more. I appreciate the authors recognizing the limitations. This was one of the reasons for suggesting showing the future weather data in fig 3. Also, consider changing the title of this paper based on this statement.

Line 315 – this is a VERY important detail that must be mentioned upfront, including the abstract. This also explains the low contribution of the SOM to future yields. As mentioned in line 287, you estimated a theoretical upper limit to yield increase in China, yet is to be seen how much of this would be realized.

Overall, this study has merit for publication, addresses an important topic, and the authors compiled numerous datasets to perform the analysis, which analysis sounds. I would like to see more discussion and framing around the interaction (GEM) than a single factor, plant density. You can't realize future yield gains due to increased plant density if the genetics are not adapted to this, even under optimum growth conditions. In the current version, the message is that increase in plant density will solve the problem, which is not correct.

Response to the reviewers' comments

We greatly appreciate the three reviewers for the valuable comments and suggestions concerning our manuscript. The comments are very helpful in improving the manuscript. We have made great efforts to revise the manuscript in response to these comments and suggestions. The details of response to each comment raised by the reviewers are listed below for your reference.

1. Responses to the First Reviewer's Comments

Reviewer #1 (Remarks to the Author):

Thank you for giving me the opportunity to review the article “Title: Can Chinese maize be self-sufficient in climate change with the 2 existing are in the 2030s? Evidences from data-driven projection 3 and field trials” in Nature communication journal.

Reviewer 1

In this manuscript, the authors aim to analyze the random forest machine-learning model to identify optimum plant density (OPDs) over China's maize areas using four different regions. The overall intention of this submission is a good one.

Thanks a lot for your kind comment.

Interested readers are expected to quickly catch your contribution, it would be better to highlight major difficulties and challenges, and your original achievements to overcome them, in a clearer way in abstract, introduction and conclusion. Explain key findings/ results in the conclusion section.

Thanks for your comment. We have revised the abstract and introduction to be clearer. The revision mode was used to show the revised parts in the manuscript for your reference. The research objective has been added at the end of introduction. Meanwhile, a paragraph has been added to the conclusions section at the end of the discussion to further highlight and explain the key findings. The added research objective in the “Introduction” section has been listed below for your reference.

We explore the potential of yield improvement through dense planting with co-regulation of management and soil quality under current and future climates through the integration of machine learning and field trials methodologies. Furthermore, the

question of whether China can be self-sufficient in maize under projected future climate change given its existing cropping areas and optimal agricultural management in the 2030s is also addressed.

The added paragraph as a conclusion is listed below for your reference.

Overall, maize yield improvement depends on complex interactions among genetics, environment and management. Based on the data-driven approach and field trials, we demonstrate that China would be self-sufficient in maize with current cropping areas in the 2030s through denser planting in combination with selecting best-suited hybrid varieties and soil improvements under future climate. The findings also indicate that high-quality soils with higher SOM could moderate the impact of climate change on OPD and thus improve maize yield. The OPD reflect the density-genetics interaction in this study. In the future, the interactions between hybrid and density could be further enhanced for yield. Our results challenge the view that grain yield have reached an attainable maximum in most global areas and provide a workable example for grain yield improvement through dense planting.

The English in the present manuscript is not of publication quality and requires major improvement. English editing is required for the whole manuscript content.

Thanks for your comment. The language has been checked by a native English speaker in this version, who is also an expert in agricultural production at the Potsdam Institute for Climate Impact Research (PIK). We also revised the “Acknowledgments” section to show the appreciation for English editing.

Minor changes and editing are required for the following sentences:

Line no-77. Hence, a critical question is whether the strategy that dense planting combined with an improved environment.

Yes, you are right. We have revised this sentence according to your suggestion in the manuscript.

The revised sentence is listed below for your reference.

Hence, a critical question is whether the strategy that dense planting combined with an improved environment such as soil-management and better cultivars in China is adequate to achieve the anticipated increase in maize yield in the coming decades?

Line no-99, the RF algorithm was (firstly) trained and tested at the nationwide level to predict

Yes, you are right. We have revised this sentence according to your suggestion in

the manuscript.

The revised sentence is listed below for your reference.

Building on a database of 2442 paired observations (yield-density points) derived from 125 studies published between 2000 and 2021 across major maize areas in China, a RF algorithm is trained and tested at the nationwide level to predict OPD as a function of environmental inputs, management and soil organic matter content (SOM).

Line no-216 condition (s)

Yes, you are right. The revised sentence is listed below for your reference.

On average, a 2.5% increase in OPDs from SOM optimization compared to simulation at current soil conditions is observed.

Line no-233 the global maize supply-demand balance. As an important indicator, planting (? Planting or what) density

Thanks. It should be “planting”.

The revised sentence is listed below for your reference.

Planting density can mediate between genotype, environment and management.

Line no-331 As a follow-up, the practices must be evaluated and modified matching high-yielding systems with optimum planting density.

Thanks. The revised sentence is listed below for your reference.

As a follow-up, practices must be evaluated and modified to match high-yielding systems with optimum planting density.

Line no-338 originated from classification and regression trees that is now increasingly used in ... ?

Thanks. We have revised this sentence and it is listed below for your reference.

As a nonparametric and ensemble learning algorithm originating from classification and regression trees, RF is now increasingly used in the crop sciences⁵⁰.

Line no- 430 Maize demand for by 2035 in China

Thank you. We have revised this sentence and it is listed below for your reference.

Maize demand by 2035 in China

Line no-394 I strongly recommended reframing the sentence above. for e.g.: Here, 402 stations where weather data and farmers' yield are available were selected to establish a regional-scale dataset.

Need to be short few sentences: (Very long sentences).

Thank you. We have revised these sentences and attach them below for your reference.

Data on farmers' planting densities and yields are included to evaluate the yield gain with dense planting. Here, we establish a dataset collected from 402 stations with available weather data, farmers' yield and planting density. Farmers' planting density (actual density) for each site is collected through surveys of the constant 24 farmers in different experimental sites in 2009–2016³². In each site, the actual yields are calculated as the average yield over the ten years from 2010 to 2019 obtained from the China Municipal Statistical Yearbook of the National Bureau of Statistics²². As the input indicators in the RF-OPD model (Fig. 6), climate data, maize phenology and soil data are essential. Daily climate data (Tmax, Tmin, precipitation, sunshine hour) for the period 2000–2020 are directly collected from the China Meteorological Administration (CMA). Daily solar radiation is calculated from observed sunshine hours using the Angstrom-Prescott equation⁵¹. Data on soil organic carbon (SOC) for each station are collected according to National Earth System Science Data Center⁵². The SOC is converted to SOM by multiplying the factor of 1.724 (SOC% × 1.724)⁵³. Observed data on maize growth (including sowing and maturity dates) from 2010 to 2018 are collected from the national agro-meteorological experiment stations across the maize cultivation areas in China⁵⁴.

Fig. 4 | Optimum density and yield from RF model and its comparison with farmers' practice and other two methods. Spatial distribution of farmers' plant density (a), OPD (b), density gap (c), farmers' yield (d), yield at OPD (e), and yield gap (f). Farmers' density was derived from Ming et al. (2017). Farmers' yield was collected from the public dataset²². Density gap equals to OPD minus farmers' density. Yield gap is defined in an analogous manner. Comparison of maize density (g) and yield (h) under different methods.

- What is Comparison in Fig 4.? IS it Comparison?

Yes, you are right. It should be comparison. We have revised the caption in the figure.

The revised figure caption is attended below for your reference.

Fig. 3 | Optimum density and yield from RF model and in comparison with farmers' practices and two other methods. Spatial distribution of farmers' planting density (a), OPD (b), density gap (c), current yield (d), yield at OPD (e), and yield gap (f). Farmers' planting density is derived from Ming et al. (2017). Current yield is

collected from the public dataset²². Density gap is equaled to OPD minus farmers' densities. Yield gap is defined in an analogous manner. Comparison of maize density (g) and yield (h) under different methods.

- What is the main reason for selecting four regions (Northeast China, NE; North China Plain, NCP; Northwest China, NW; Southwest China, SW)? What are the advantages of adopting these locations over others in this study? How will this affect the results? More details should be furnished.

Generally, maize agro-ecological regions in China could be divided into four regions due to climate, management and maize pattern from north to south: Northeast China (NE), North China Plain (NCP), Northwest China (NW) and Southwest China (SW) (Liu et al., 2017; Meng et al., 2018). Accordingly, the research region was also divided into these four regions (NE, NCP, NW and SW). In this study, the research region covered 91% of national maize sowing area and produced roughly 90% of national output (National Bureau of Statistics, 2022).

Because of the diversity of climates and managements among four regions, the study area severed as an excellent laboratory to investigate maize production. The region extended from southern tropical and sub-tropical systems (SW) at low latitudes to cool-temperate systems (NE) at high latitudes. Meanwhile, the maize system was quite different among regions. For example, the spring maize was the dominant maize system in NE and SW. In NCP, summer maize followed winter wheat was the dominant system in the NCP.

We have furnished the above information in a separate paragraph according to the comments from you and the 2nd reviewer.

The revised section was listed below for your reference.

Study area

China is the second major producer of maize, constituting one-fifth of global production (Supplementary Fig. 5). Generally, maize agro-ecological regions in China could be divided into four regions based on climate, management and maize cropping pattern from north to south³⁵: Northeast China (NE), North China Plain (NCP), Northwest China (NW) and Southwest China (SW). The study area in Fig. 1a covered 91% of the national maize cropping areas and produced roughly 90% of national output²². Because of the diversity of climate and management among these four regions, the study area serves as an excellent laboratory for further improving maize yields.

The related references are also listed below for your reference.

Liu, B., Chen, X., Meng, Q., Yang, H. & van Wart, J. Estimating maize yield potential and yield gap with agro-climatic zones in China Distinguish irrigated and rainfed conditions. *Agric. For. Meteorol.* 239, 108-117 (2017).

Meng, Q., Cui, Z., Yang, H., Zhang, F. & Chen, X. Establishing High-Yielding Maize System for Sustainable Intensification in China. *Adv. Agron.* 148, 85-109 (2018).

National Bureau of Statistics. *China Municipal Statistical Yearbook*
<https://data.stats.gov.cn/>. Accessed 1 May 2022.

Modeling flowcharts for the developed methodology shall be carefully added to the revised manuscript.

Yes, you are right. We have added the flowchart in the main text as you requested. This figure is referred as Fig. 6 in the new manuscript and it is attached below for your reference.

Fig. 6 | Framework of the procedure for mapping regional optimum plant density (OPD) and yield at OPD (Yield_{OPD}). T_{max}: daily maximum temperature. T_{min}: daily minimum temperature. GDD: growing degree days (10–30 °C). SOM: soil organic matter. T_{max} and T_{min} is averaged during maize growing season, and solar radiation, precipitation and GDD is the sum of growing season.

Make sure that all references within the style of the journal.

Thank you very much. We have checked and revised the reference according to the format request from the journal.

2. Responses to the Second Reviewer's Comments

Reviewer #2 (Remarks to the Author):

The manuscript entitled "Can Chinese maize be self-sufficient in climate change with the existing area in the 2030s? Evidences from data-driven projection and field trials" addressed a very important issue on the global largest crop in total production such as maize production under climate change. I was very impressed with this innovative study because of the similar results from two independent methodologies. While a lot of current researches have been applying the machine learning method for crop production prediction, there are rarely field based observations, especially across wide ecological regions. This study will fill this research gap and supply an excellent example for the combination of two methodologies. The results showed a 55% yield improvement with dense planting in climate change in 2030s in climate change, and the author found the self-sufficient supply for Chinese maize with existing arable land. Interestingly, yield gains from soil improvement with high SOC was more than the negative impact by climate change in next decades. Overall, the result of the manuscript is an important scientific advancement and finding. Before the manuscript could be accepted for publication, it needs a major revision for further improvement, especially in the methods section for readers to early follow and understand.

Thank you very much for your kind comments.

We have revised the manuscript according to your comments and suggestions, especially in the methods section. The point-to-point response is appended below for your reference.

The following comments and suggestions are provided for your consideration.

1. Lines 27-29. "we estimated a 55% yield improvement through dense planting together with soil improvement compared with the historical trend in the climate under high Shared Socio Economic Pathways", this sentence should be revised. After reading the whole manuscript, I finally understand this sentence. Please revise, especially with the followed sentence "Maize yield was estimated to roughly doubled than current farmers with rational dense planting". It would be better to distinguish the two yield improvements in different period.

Yes, we have revised the two sentences according your comments.

The revised sentences are listed below for your reference.

Current maize yield with rational denser planting and optimal management would be roughly doubled than present farmers. In the 2030s, we estimate a 52% yield improvement through dense planting together with soil improvement compared with the historical trend in the climate, under a high-end climate forcing Shared Socio-Economic Pathway (SSP585).

2. At the end of the introduction, the objective of the study should be described further. This will help the readers to understand the whole story of the manuscript.

Yes, you are right. We have added the objective at the end of the introduction.

The added part is listed below for your reference.

We explore the potential of yield improvement through dense planting in combination with best-suited hybrid varieties and soil improvements under current and future climates through the integration of machine learning and field trials methodologies. Furthermore, the question of whether China can be self-sufficient in maize under projected future climate change given its existing cropping areas and optimal agricultural management in the 2030s is also addressed.

3. Line 53-54 “The gain of crop yield relies on complex interactions among genotype, environment (including climate and soil conditions), and management.” Please add the reference for this statement.

Thanks for your comment. We have added a reference for this sentence.

The revised sentence is listed below for your reference.

Crop yield gains rely on complex interactions between genotypes, environmental factors (including climate and soil conditions), and agricultural management⁸.

The added reference is also listed below for your reference.

8. *Duvick, D. Genetic progress in yield of United States maize (Zea mays L.). Maydica 50, 193 (2005).*

4. Line 66 Fig.1 is not the main results. It's just a background for maize production between China and US. I suggest to remove Fig.1 to the supplementary information.

Yes, you are right. We have removed it to the supplementary information in the manuscript (Supplementary Fig. 1).

5. Line 101-102 This looks like the discussion of your study, I think the author should emphasize your research objective in this section as I mentioned above.

Yes, you are right. We have deleted this sentence and added the research

objectives in this section.

The added objectives are listed below for your reference.

We explore the potential of yield improvement through dense planting in combination with best-suited hybrid varieties and soil improvements under current and future climates through the integration of machine learning and field trials methodologies. Furthermore, the question of whether China can be self-sufficient in maize under projected future climate change given its existing cropping areas and optimal agricultural management in the 2030s is also addressed.

6. Line 247 What reasons may cause the average OPD from our RF model is lower than that of the North America region.

Thanks for your comment. OPD is determined by complex interactions among genotype, environment, and management factors (Assefa et al. 2018). To our knowledge, there are several reasons for the lower OPD in China than that of the North America region.

Initially, the crop systems are quite different between the two regions. Maize monoculture or rotated with soybean is the major system in maize production in the North America. If maize is sown, it is one harvest in one year as spring maize (Seifert et al., 2017). In contrast, in China maize is produced in diverse crop systems (Tao et al., 2016). For example, the summer maize and winter wheat rotation with two harvests in one year is the main crop system in the North China Plain. Due to the relative shorter growth season, the OPD of summer maize is generally lower than spring maize.

Furthermore, the better weather, especially the increased solar radiation experienced in the North America region during past decades, contributes to higher OPD (Tollenaar et al., 2017). In contrast, the solar radiation in the major maize growing area in China (e.g., North China Plain) during the maize season was decreased in the past decades (Meng et al., 2020).

In addition, the soil conditions are different between two regions. A recent study found that the average SOM was 29 g kg⁻¹ in two important maize regions in North America region such as Wisconsin and Minnesota (Oldfield et al., 2022). The SOM concentration varied from 14 to 23 g kg⁻¹ with an average of 19 g kg⁻¹

across the four maize growing regions in China (NESSDC-SSC, 2022).

The related references are listed below for your reference.

- Assefa, Y. et al. Analysis of long-term study indicates both agronomic optimal plant density and increase maize yield per plant contributed to yield gain. Sci. Rep. 8, 1-11 (2018).*
- Meng, Q., Liu, B., Yang, H. & Chen, X. Solar dimming decreased maize yield potential on the North China Plain. Food Energy Secur. 9, e235 (2020).*
- NESSDC-SSC, Soil SubCenter, National Earth System Science Data Center, National Science & Technology Infrastructure of China. China High-resolution National Soil Information Grid Basic Attribute Dataset (2010–2018) <http://soil.geodata.cn>. Accessed 17 January 2022.*
- Oldfield, E. E. et al. Positive associations of soil organic matter and crop yields across a regional network of working farms. Soil Sci. Soc. Am. J. 86, 384-397 (2022).*
- Seifert, C. A., Roberts, M. J. & Lobell, D. B. Continuous Corn and Soybean Yield Penalties across Hundreds of Thousands of Fields. Agron. J. 109, 541-548 (2017).*
- Tao, F. et al. Historical data provide new insights into response and adaptation of maize production systems to climate change/variability in China. Field Crops Res. 185, 1-11 (2016).*
- Tollenaar, M., Fridgen, J., Tyagi, P., Stackhouse Jr, P. W. & Kumudini, S. The contribution of solar brightening to the US maize yield trend. Nat. Clim. Change 7, 275-278 (2017).*

7. Line 249-L253 Machine learning techniques have been successfully implemented in agricultural production to investigate various agronomic indicators (e.g., crop yield), Why do you choose the machine learning techniques? What are its advantages? It would be good to add some discussion here.

Thanks for your suggestion. We have added the advantages of machine learning technique in this part.

The added sentences are listed below for your reference.

As a popular decision-tree-based ensemble machine learning algorithm, RF can handle nonlinear effects and complex interactions among variables²⁷. Through the implementation as an RF model, the OPD projection is data-driven and does not rely on pre-specified equations or functional form.

The new added reference is listed below for your reference.

27. *Liakos, K. G., Busato, P., Moshou, D., Pearson, S. & Bochtis, D. Machine learning in agriculture: A review. Sensors 18, 2674 (2018).*

8. Line 328-331 It should be stated here how this database was built, instead of stating

China is the second major producer of maize, constituting one-fifth of global production, the study region had better use a separate paragraph description.

Yes, you are right. We have added a subtitle together with a separated paragraph for description of the study region.

The revised section is listed below for your reference.

Study area

China is the second major producer of maize, constituting one-fifth of global production (Supplementary Fig. 5). Generally, maize agro-ecological regions in China could be divided into four regions based on climate, management and maize cropping pattern from north to south³⁵: Northeast China (NE), North China Plain (NCP), Northwest China (NW) and Southwest China (SW). The study area in Fig. 1a covered 91% of the national maize cropping areas and produced roughly 90% of national output²². Because of the diversity of climate and management among these four regions, the study area serves as an excellent laboratory for further improving maize yields.

9. Line 335-336 Case writing should be standardized.

Yes, you are right. We have revised this part in the manuscript.

The revised section is shown below for your reference.

The literature search was performed using the China National Knowledge Infrastructure and the Web of Science for relevant papers published between January 2000 and October 2021 using the following keywords: “density” AND “yield*” AND (“maize*” OR “corn*”) AND (“China*” OR “Chinese*”).*

10. Line 363-364 “we found a linear relationship between Yield_{OPD} and OPD across all regions (Fig. 2d).” This is the result of the article and should not appear in materials and methods.

Thanks. We have added this information in the caption of Fig. 1.

The revised section is listed below for your reference.

*Fig. 1 | The Random Forest (RF) model development. a, Study area and locations of experimental sites found in the literature. b, Data distributions of OPDs and Yield_{OPD} estimated from literature data with yield-density quadratic model. c, Comparisons of the estimated OPD by RF model with observed OPD in 448 observations. The dashed line represents the 15% error line. d, Linear-model for the relationship between OPD and Yield_{OPD}. We find a similar performance in linear and quadratic fits for the relationship between the Yield_{OPD} and OPD and the linear nature is applied. Shaded areas show 95% confidence interval. The asterisks indicate the statistical significance of the effect (*P < 0.05; ** P < 0.01; ***P < 0.001).*

11. Line 393-415 should be improved and expressed clearly, for example, the prediction was station-level, however, the Supplementary Figure 3 were county-scale, which is confused.

Yes, you are right. Our OPD projection is based on a station-level dataset. We revised the “OPD projection under current and future climate data” section in the Methods according to your suggestions. We also revised the label of “county-scale” with “station-level” in Fig. 6.

The revised section is listed below for your reference.

Data on farmers' planting densities and yields are included to evaluate the yield gain with dense planting. Here, we establish a dataset collected from 402 stations with available weather data, farmers' yield and planting density. Farmers' planting density (actual density) for each site is collected through surveys of the constant 24 farmers in different experimental sites in 2009–2016³². In each site, the actual yields are calculated as the average yield over the ten years from 2010 to 2019 obtained from the China Municipal Statistical Yearbook of the National Bureau of Statistics²². As the input indicators in the RF-OPD model (Fig. 6), climate data, maize phenology and soil data are essential. Daily climate data (Tmax, Tmin, precipitation, sunshine hour) for the period 2000–2020 are directly collected from the China Meteorological Administration (CMA). Daily solar radiation is calculated from observed sunshine hours using the Angstrom-PreScott equation⁵¹. Data on soil organic carbon (SOC) for each station are collected according to National Earth System Science Data Center⁵². The SOC is converted to SOM by multiplying the factor of 1.724 (SOC% × 1.724)⁵³. Observed data on maize growth (including sowing and maturity dates) from 2010 to 2018 are collected from the national agro-meteorological experiment stations across the maize cultivation areas in China⁵⁴.

To predict OPD under climate warming, future scenario data is obtained from Global Climate Models (GCMs), which was contributed by the World Climate Research Program (WCRP) of Coupled Model Inter-comparison Project Phase 6 (CMIP6, <https://esgf-node.llnl.gov/search/cmip6>). Daily weather data (daily temperature, precipitation, and solar radiation) from 2010 to 2039 for 22 GCMs (Supplementary Table 5) under a high radiative forcing Shared Socio-Economic Pathway (SSP585) was downscaled from monthly gridded data using the statistical downscaling model NWA1-WG⁵⁵. We simulated OPD and YieldOPD for the current period 2010-2019 (abbreviated as 2010s) and future period 2030-2039 (abbreviated as 2030s) with the framework in Fig. 6.

The revised Fig.6 is also listed below for you reference.

Fig. 6 | Framework of the procedure for mapping regional optimum plant density (OPD) and yield at OPD (Yield_{OPD}). Tmax: daily maximum temperature. Tmin: daily minimum temperature. GDD: growing degree days (10–30 °C). SOM: soil organic matter. Tmax and Tmin is averaged during maize growing season, and solar radiation, precipitation and GDD is the sum of growing season.

12. Lines 397-399. “The average yield was calculated as the average yield over the ten years from 2010 to 2019 obtained from the China Municipal Statistical Yearbook of the National Bureau of Statistics⁴¹ to represent actual farmers’ yield in each site”, why not directly use the yield of 402 stations, but use the yield in the statistical yearbook.

Thanks for your comments. In our study, data on farmers’ yield was included to evaluate the yield gain with applications of dense planting. Farmers’ plant density for each site was collected through surveys of the constant 24 farmers at different experimental sites in 2009–2016 (Ming et al., 2017). However, there was a lack of reports on farmers’ yield in this database. Therefore, we further collected yield data from the China Municipal Statistical Yearbook of the National Bureau of Statistics (National Bureau of Statistics, 2022) as the actual farmers’ yield in each site.

The reference mentioned is listed below for your reference.

Ming, B. et al. Changes of maize planting density in China. Sci. Agric. Sin. 50, 1960-

1972 (2017).
National Bureau of Statistics (NBS). China Municipal Statistical Yearbook
<https://data.stats.gov.cn/>. Accessed 1 May 2022.

13. Line 429 “Maize demand for by 2035 in China”, should be “Maize demand by 2035 in China”

Yes, you are right. We have revised this sentence and listed below for your reference.

Maize demand by 2035 in China

14. Line 436-437 please show the data source for the population. If possible, please add a reference.

Thanks for your comment. We have added the data source for the population and revised the related sentence in the manuscript.

China’s population is predicted to increase from 1.428 billion in 2021 to 1.434 billion in 2035⁵⁸, resulting in 292 Mt of maize demand by 2035.

The reference of the data source is listed below.

58. *Department of Economic and Social Affairs. World Urbanization Prospects 2018* <https://population.un.org/wup/DataQuery>. Accessed 23 Aug 2022.

3. Responses to the Third Reviewer’s Comments

Reviewer #3 (Remarks to the Author):

The authors estimated a 55% maize yield increase in China by 2035 if two conditions are met: increased plant density and soil organic matter improvements. This estimate was derived from a machine learning model, previously calibrated and tested for China.

Thanks a lot for your kind comments.

While I do agree the yields will continue to increase by 2035, I am skeptical about the 55% estimate for three reasons:

1) plant density changes occur concurrently with changes in maize hybrids, and you can’t separate density from genetics with this dataset. You modeled the density-genetics interaction, not just the plant density. The citation “8” refers to a well-irrigated region (Nebraska), not reflecting the entire Corn Belt;

Yes, you are right. The modeled density in this study was based on the density-

genetics interaction, not just the plant density as you mentioned. The density-genetics interaction was based on the existing data with current hybrids. In future, grain yield would be further enhanced with major breakthroughs in genetic improvement with new hybrids, such as more compact plant shape with higher resistance for dense planting. In order to make this information clearer, we have added a paragraph at the end of the “Discussion” section and two sentences in “introduction” section.

To separate the contribution of density, genetics and density*genetics, we have further analyzed the field data according to Tollenaar and Lee (2002) and Ma et al. (2015). Our results with 87 field experiments showed that maize yield was improved by 21% with the density-genetics interaction compared to the control. The yield grain from genetics and density improvement was 5.9% and 7.3%, respectively. The density*genetics interaction contributed 7.4% of yield improvement. We have added a Supplementary Figure to show this result (Supplementary Fig. 3).

The newly added Supplementary Fig. 3 was also listed below for your reference.

Supplementary Figure 3. Yield improvements from genetics, density and interaction of the both. a, The yield improvement of maize in field observations (87-trials) from the contribution of the genetic (DF), planting density (EF) and density-genetic interaction (CE)^{5,6}. A (or D) represented the control: using the local maize hybrid and farmers’ density. B was the local hybrid grown in a high-density. C represented the optimum treatment (OT): OPD with high-yielding maize hybrids. E was the yield of high-

yielding maize hybrids with increased density. F indicated the high-yielding hybrids at the farmers' planting density. b, the plant morphology of a control hybrid (Zhengdan958) and high-yielding hybrid (Jingnongke728).

The related references are listed below for your reference.

Tollenaar, M. & Lee, E. Yield potential, yield stability and stress tolerance in maize. Field Crops Res. 75, 161-169 (2002).
Ma, D. et al. Genetic contribution to maize yield gain among different locations in China. Maydica 60, 1-8 (2015).

We have also added the description for the above figure in the manuscript. The added information is listed below for your reference.

*Furthermore, the yield gain from genetic and density improvements is 5.9% and 7.3%, respectively (Supplementary Fig. 3). The density*genetics interaction contributes 7.4% of yield improvement.*

The new added paragraph at the end of the “Discussion” section is also listed below for your reference.

Overall, maize yield improvement depends on complex interactions among genetics, environment and management. Based on the data-driven approach and field trials, we demonstrate that China would be self-sufficient in maize with current cropping areas in the 2030s through denser planting in combination with selecting best-suited hybrid varieties and soil improvements under future climate. The findings also indicate that high-quality soils with higher SOM could moderate the impact of climate change on OPD and thus improve maize yield. The OPD reflect the density-genetics interaction in this study. In the future, the interactions between hybrid and density could be further enhanced for yield. Our results challenge the view that grain yield have reached an attainable maximum in most global areas and provide a workable example for grain yield improvement through dense planting.

The added part at the end of “Introduction” section is also listed below for your reference.

We explore the potential of yield improvement through dense planting in combination with best-suited hybrid varieties and soil improvements under current and future climates through the integration of machine learning and field trials methodologies. Furthermore, the question of whether China can be self-sufficient in maize under projected future climate change given its existing cropping areas and optimal agricultural management in the 2030s is also addressed.

The statement about citation “8” has also been revised. The revised sentence is

listed below for your reference.

Recent work in three irrigated maize regions (Lower Niobrara, Tri-Basin, and Upper Big Blue) in Nebraska illustrates that climate trends and agronomic improvements, not genetic improvements, underpin recent maize yield gains⁹.

2) to realize future yield improvements due to crop improvement (density x genetics changes) other management aspects should be altered in the production system to support the new potential (irrigation, fertilization). This is somewhat discussed in the paper, but it is not highlighted enough;

Yes, you are right. Yield gain resulted from improved density and hybrids would be supported by the enhanced management strategy (e.g., irrigation, fertilization). We have added one paragraph to highlight the importance of management in the “Discussion” section.

The new added paragraph is listed below for your reference.

The achievement of modeled OPDs would need to be supported by enhanced management strategies (e.g., irrigation and fertilization). Increasing planting density increases plant-to-plant competition in high-yielding systems⁴² and induces greater sensitivity to drought in maize³⁴. Drought is one of most important limitations to yield gains in 40% of maize area in China²¹ and irrigation should be considered in the intensification of maize production systems, which has been shown to be an effective management practice for improving yields⁴³. Meanwhile, nitrogen fertilizer applied at the critical stages for high-planting density is required for optimal maize yield⁴⁴. To this end, an integrated crop-soil system approach in management will be required in the future to support high-yielding maize system with dense planting⁴⁵.

The new added references also are listed below for your reference.

42. Meng, Q., Cui, Z., Yang, H., Zhang, F. & Chen, X. *Establishing High-Yielding Maize System for Sustainable Intensification in China. Adv. Agron. 148, 85-109 (2018).*
43. Zhu, P. & Burney, J. *Untangling irrigation effects on maize water and heat stress alleviation using satellite data. Hydrol. Earth Syst. Sci. 26, 827-840 (2022).*
44. Ciampitti, I. A. & Vyn, T. J. *Physiological perspectives of changes over time in maize yield dependency on nitrogen uptake and associated nitrogen efficiencies: A review. Field Crops Res. 133, 48-67 (2012).*
45. Chen, X. P. et al. *Integrated soil-crop system management for food security. Proc. Natl Acad. Sci. USA 108, 6399-6404 (2011).*

3) how realistic is it to expect soil improvements within a 10-yr period? Besides biochar application, I don't see any other way to increase soil organic matter within such a short time period.

Thanks for your comments. Achieving soil improvement such as increasing soil organic matter (SOM) is a great challenge worldwide. There are still some possibilities in Chinese maize production.

The current value of SOM is relative lower in China compared with the US in maize area. A recent study found that the average SOM content was 29 g kg⁻¹ in two important maize regions in North America region such as Wisconsin and Minnesota (Oldfield et al., 2022). The average SOM content was 19 g kg⁻¹ in major maize growing regions in China (NESSDC-SSC, 2022). The lower base value of SOM content in China offers the high possibility of rapid improvement in the relative short-term period.

According to a 9-yr (2007-2016) field experiment with optimized N management and straw return in a summer maize-winter wheat rotation system in North China Plain, the soil organic carbon stock in the upper 0-30 cm soil layer was improved by 14% (Pan et al., 2019). In the past thirty years (1980-2011), China's croplands experienced a 15% increase of the average soil organic carbon stock in the topsoil (0–20 cm) (Zhao et al., 2018). This national level soil organic carbon improvement resulted from increased organic inputs: root inputs (4.6%), straw return (6.1%), and other C inputs (4.6%).

The mentioned references are listed below for your reference.

Oldfield, E. E. et al. Positive associations of soil organic matter and crop yields across a regional network of working farms. *Soil Sci. Soc. Am. J.* 86, 384-397 (2022).

Pan, J., Zhang, L., He, X., Chen, X. & Cui, Z. Long-term optimization of crop yield while concurrently improving soil quality. *Land Degrad Dev* 30, 897-909 (2019).

Soil Survey Staff, Natural Resources Conservation Service, United States Department of Agriculture. Web Soil Survey. Available online at <https://websoilsurvey.nrcs.usda.gov/>

Zhao, Y. et al. Economics- and policy-driven organic carbon input enhancement dominates soil organic carbon accumulation in Chinese croplands. *Proc. Natl Acad. Sci. USA* 115, 4045-4050 (2018).

Line 59 – related to my earlier concern, the authors cite “11” which refers to optimum plant density increase but failed to report a key message from that paper that “plant density contribution to maize yield gain ranged from 8.5% to 17%”.

Yes, you are right. We have revised the sentence as follows:

In North America, optimum plant density (OPD) increased at a rate of 700 plants per hectare per year during 1987–2016¹². Plant density contribution to maize yield gain ranges from 8.5% to 17%.

Line 72 and many other places – explain acronyms in the first mention.

Thank you very much. The acronym of “OPD” was mentioned earlier and explained in the first mention.

The sentence is listed below for your reference.

In North America, optimum plant density (OPD) increased at a rate of 700 plants per hectare per year during 1987–2016¹².

According to your suggestion, we also checked the whole manuscript and make sure that acronyms are explained in the first mention.

Fig 1. Interesting presentation. FYI, according to USDA-NASS statistics, plant densities have been stabilized (or the increase has slowed down) over the last 5 years in the US Corn Belt, yet the grain yield continues to increase.

Yes, you are right. According to the data from USDA-NASS, planting density in the US was stabilized over the past five years (2018-2022) (USDA, 2023). The density averaged 7.2×10^4 plants ha⁻¹ with a range from 7.0 to 7.3×10^4 plants ha⁻¹ although the grain yield continued to increase. In most states, the density may already be close to the optimal level of the current hybrids in practice (Assefa et al., 2018). Under this condition, the enhanced management such as irrigation and fertilization would contribute largely for the yield improvement. For example, a survey data collected from a subset of 268 farmers across three regions in the Nebraska showed the applied N fertilizer contributed 50 kg ha⁻¹ yr⁻¹ of maize yield gain while it was only 28 kg ha⁻¹ yr⁻¹ from seeding rate between 2005 and 2018 (Rizzo et al., 2022).

There are substantial differences in maize yield and density between US and China. The first step was to determine OPD for yield improvement in China. The

figure with the comparison of grain yield and planting density would great help to understand the situation of maize production in the world's top two maize-producing countries. In addition, the figure has already been moved to the “Supplementary Information” (Supplementary Fig. 1) as requested by the 2nd Reviewer.

The reference mentioned above is listed below for your reference.

Assefa, Y. et al. Analysis of long term study indicates both agronomic optimal plant density and increase maize yield per plant contributed to yield gain. *Sci. Rep.* 8, 1-11 (2018).

Rizzo, G. et al. Climate and agronomy, not genetics, underpin recent maize yield gains in favorable environments. *Proc. Natl Acad. Sci. USA* 119, e2113629119 (2022).

United States Department of Agriculture. National Agricultural Statistics Service https://quickstats.nass.usda.gov/Quick_Stats/. Accessed 7 April 2022.

Line 113. Is GDD for the growing season? If yes, I assume you have planting date data and if that is the case, I wonder about the role of planting date on the optimum plant density.

Yes, you are right. GDD was included as a cultivar indicator, reflecting the thermal time for the growing season. We have further analyzed the role of planting date and the result is shown for your reference.

Importance of sowing date on OPD (a) and relationships between sowing date and OPD (b). NE: Northeast China, NCP: North China Plain, NW: Northwest China, SW: Southwest China. The symbols + and - indicated positive and negative effects of the variable on OPD, respectively. The asterisks indicate the statistical significance of the effect (* $P < 0.05$; ** $P < 0.01$; *** $P < 0.001$).

Although there are some significant negative relationships between OPD and the

sowing date, the RF-OPD model was not significantly improved according to evaluations from the R^2 , $RMSE$ and $RRMSE$ (figure below). Thus, the sowing date was not taken into accounts into the model.

Comparisons of the estimated OPD by RF model with observed OPD in 448 observations. Note: the RF model included seven indicators (i.e., T_{min} , T_{max} , Rad_n , $Prec$, GDD , SOM and sowing date). The dashed line represents the 15% error line. The asterisks indicate the statistical significance of the effect ($P < 0.05$; ** $P < 0.01$; *** $P < 0.001$).*

Fig 2d, the linear nature of the relationship is interesting. Would be good to expand/explain this.

Thanks for your comment and suggestion. We find a similar performance in linear and quadratic fits for the relationship between the YieldOPD and OPD and the linear nature is applied in this study. We have added this information in the caption of Fig. 1.

The revised section is listed below for your reference.

Fig. 1 | The Random Forest (RF) model development. a, Study area and locations of experimental sites found in the literature. b, Data distributions of OPDs and YieldOPD estimated from literature data with yield-density quadratic model. c, Comparisons of the estimated OPD by RF model with observed OPD in 448 observations. The dashed line represents the 15% error line. d, Linear-model for the relationship between OPD and YieldOPD. We find a similar performance in linear and quadratic fits for the relationship between the YieldOPD and OPD and the linear nature is applied. Shaded areas show 95% confidence interval. The asterisks indicate the statistical significance of the effect ($P < 0.05$; ** $P < 0.01$; *** $P < 0.001$).*

Line 133 change improvement to increase.

Yes, you are right. We have revised it as follows:

T_{min} significantly decreases OPD from 0.16 to 0.51 × 10⁴ plants ha⁻¹ per 1 °C increase while Radn increase 1.00 to 2.00 × 10⁴ plants ha⁻¹ per 1000 MJ m⁻² increase.

Fig 3. Please specify years (time period) of observations, I assume this is for the past years. Also, update the graph by adding the future weather years, probably in the supplemental materials?

Yes, you are right. The observations were derived from literatures in 2000-2021.

We revised the figure caption as follows:

Fig. 2 | Importance of variables influencing OPD and relationships between drivers and OPD for each region based on observations from the literatures in 2000-2021.

We have added a new Supplementary Fig. 2 with the future weather in the Supplementary Information. Meanwhile, a sentence is added in the text to conclude the figure information.

The added sentence is listed below for your reference.

In the future, the influence of T_{max} on OPD is projected to be strengthened (Supplementary Fig. 2).

The new added figure (Supplementary Fig. 2) is also attached here for your reference.

Supplementary Figure 2. Relationships between OPD and climate variables, daily minimum temperature (a), daily maximum temperature (b), accumulate solar radiation (c) and precipitation (d) in maize growing season. Observations based on the literatures (2000-2021) and RF projection in the 2030s. The study area was divided in to four regions: Northeast China (NE), North China Plain (NCP), Northwest China (NW) and Southwest China (SW). The asterisks indicate the statistical significance of the effect ($P < 0.05$; ** $P < 0.01$; *** $P < 0.001$).*

Why the current plant density is low in China? Must be a reason. Water limitation?

Thanks for your question. In order to show the reason for the current low plant density, we have added a Supplementary Table (Supplementary Table 5) in the manuscript. Meanwhile, we also summarized the limiting factor in each region in the “Discussion” section.

The new added Supplementary Table 5 is shown below for your reference.

Supplementary table 5. Factors that limit farmers' planting density increases among the four regions. Northeast China (NE), North China Plain (NCP), Northwest China (NW), and Southwest China (SW).

Region	Limitations						References
	Maximum temperature	Minimum temperature	Solar radiation	Irrigation	Soil	Lodging	
NE		√				√	Ming et al., 2017; Li & Wang, 2008; Lobell et al., 2014; Liu et al., 2017; Wu et al., 2019; Meng et al., 2020; Bu et al., 2015; Li et al., 2012; Xue et al., 2016
NCP	√		√		√	√	
NW				√	√	√	
SW	√				√	√	

The related references in “Supplementary Table 5” are listed below for your reference.

- Bu, L. et al. The effect of adapting cultivars on the water use efficiency of dryland maize (Zea mays L.) in northwestern China. Agric. Water Manage. 148, 1-9 (2015).*
- Li, J., Lammerts van Bueren, E. T., Jiggins, J. & Leeuwis, C. Farmers' adoption of maize (Zea mays L.) hybrids and the persistence of landraces in Southwest China: implications for policy and breeding. Genet. Resour. Crop Evol. 59, 1147-1160 (2012).*
- Li, S. & Wang, C. Analysis on change of production and factors promoting yield increase of corn in China. J. Maize Sci. 4, 26-30 (2008).*
- Liu, B., Chen, X., Meng, Q., Yang, H. & van Wart, J. Estimating maize yield potential and yield gap with agro-climatic zones in China Distinguish irrigated and rainfed conditions. Agric. For. Meteorol. 239, 108-117 (2017).*
- Lobell, D. B. et al. Greater sensitivity to drought accompanies maize yield increase in the US Midwest. Science 344, 516-519 (2014).*
- Meng, Q., Liu, B., Yang, H. & Chen, X. Solar dimming decreased maize yield potential on the North China Plain. Food Energy Secur. 9, e235 (2020).*
- Ming, B. et al. Changes of maize planting density in China. Sci. Agric. Sin. 50, 1960-1972 (2017).*
- Wu, A., Hammer, G. L., Doherty, A., von Caemmerer, S. & Farquhar, G. D. Quantifying impacts of enhancing photosynthesis on crop yield. Nat. Plants 5, 380-388 (2019).*
- Xue, J. et al. Effects of light intensity within the canopy on maize lodging. Field Crops Res. 188, 133-141 (2016).*

Meanwhile, we also revised the paragraph in the “Discussion” section to

summarized the limiting factors for density improvement. The revised paragraph is listed below for your reference.

Although our results are theoretically feasible for farmers adoption and confirmed by multi-field trials (Fig. 4), several factors limit farmers' planting density decisions such as soil water, air temperature and solar radiation (Supplementary Table 5)^{32, 33}. Increasing temperature will increase atmospheric water demand which could lead to drought stress from increased vapor pressure deficit (VPD), subsequently reducing plant density and decreasing yield across most maize areas³⁴. Low temperatures at the earlier growth stage together with water stress in the NE is a critical limitation for maize density improvement³⁵. Solar radiation, which influences photosynthesis in plant leaves as the energy source in crop production³⁶, has decreased over the past decades in the NCP³⁷, potentially limiting further increases in plant density and maize yield. A lack of irrigation in the NW and diverse landforms and ecosystems in SW were the major restrictions to increasing plant density in these regions^{38, 39}. In addition, increased risks of lodging arising from greater plant densities influence farmers' sowing density selections⁴⁰. Co-efforts from breeders and producers will determine further increases in China's maize planting density and yield. An ideal plant architecture with an appropriate canopy structure that intercepts more solar radiation is crucial to dense planting and achieving high yields³¹. Tian et al.⁴¹ report that upright plant architecture in modern hybrids provides opportunities for dense planting, providing new insights into high-density-yield maize breeding. As a follow-up, practices must be evaluated and modified to match high-yielding systems with optimum planting density.

Fig 6 great presentation.

Thanks a lot for your kind comment.

Line 287 – 289 can't agree more. I appreciate the authors recognizing the limitations. This was one of the reasons for suggesting showing the future weather data in fig 3. Also, consider changing the title of this paper based on this statement.

Thanks for your comment. We have added the future weather data in the “Supplementary Fig. 2” and attached it in the above response.

The title of the manuscript has been revised as follows:

Can China be self-sufficient in maize in the 2030s under current areas and climate change with optimal management? Evidence from data-driven projections and field trials

Line 315 – this is a VERY important detail that must be mentioned upfront, including the abstract. This also explains the low contribution of the SOM to future yields. As

mentioned in line 287, you estimated a theoretical upper limit to yield increase in China, yet is to be seen how much of this would be realized.

Yes, you are right. We have added the related information in the title, abstract and introduction.

The revised sections are listed below for your reference.

***The title:** Can China be self-sufficient in maize in the 2030s under current areas and climate change with optimal management? Evidence from data-driven projections and field trials*

***The abstract:** Current maize yield with rational denser planting and optimal management would be roughly doubled than present farmers.*

***The end of introduction:** Furthermore, the question of whether China can be self-sufficient in maize under projected future climate change given its existing cropping areas and optimal agricultural management in the 2030s is also addressed.*

Overall, this study has merit for publication, addresses an important topic, and the authors compiled numerous datasets to perform the analysis, which analysis sounds. I would like to see more discussion and framing around the interaction (GEM) than a single factor, plant density. You can't realize future yield gains due to increased plant density if the genetics are not adapted to this, even under optimum growth conditions. In the current version, the message is that increase in plant density will solve the problem, which is not correct.

Thanks a lot. We highly appreciated your positive comments on the significance of this study. The modeled density in this study was based on the density-genetics interaction, not just the plant density as you mentioned. The density-genetics interaction was based on the existing data with current hybrids. In future, grain yield would be further enhanced with major breakthroughs in genetic improvement with new hybrids, such as more compact plant shape with higher resistance for dense planting. In order to make this information clearer, we have added a paragraph at the end of the "Discussion" section and two sentences in "introduction" section.

To separate the contribution of density, genetics and density*genetics, we have further analyzed the field data according to Tollenaar and Lee (2002) and Ma et al. (2015). Our results with 87 field experiments showed that maize yield was improved by 21% with the density-genetics interaction compared to the control.

The yield grain from genetics and density improvement was 5.9% and 7.3%, respectively. The density*genetics interaction contributed 7.4% of yield improvement. We have added a Supplementary Figure to show this result (Supplementary Fig. 3).

The newly added Supplementary Fig. 3 was also listed below for your reference.

Supplementary Figure 3. Yield improvements from genetics, density and interaction of the both. a, The yield improvement of maize in field observations (87-trials) from the contribution of the genetic (DF), planting density (EF) and density-genetic interaction (CE)^{5,6}. A (or D) represented the control: using the local maize hybrid and farmers' density. B was the local hybrid grown in a high-density. C represented the optimum treatment (OT): OPD with high-yielding maize hybrids. E was the yield of high-yielding maize hybrids with increased density. F indicated the high-yielding hybrids at the farmers' planting density. b, the plant morphology of a control hybrid (Zhengdan958) and high-yielding hybrid (Jingnongke728).

The related references are listed below for your reference.

Tollenaar, M. & Lee, E. Yield potential, yield stability and stress tolerance in maize.

Field Crops Res. 75, 161-169 (2002).

Ma, D. et al. Genetic contribution to maize yield gain among different locations in

China. *Maydica* 60, 1-8 (2015).

We have also added the description for the above figure in the manuscript. The added information is listed below for your reference.

Furthermore, the yield gain from genetic and density improvements is 5.9% and 7.3%, respectively (Supplementary Fig. 3). The density*genetics interaction contributes 7.4% of yield improvement.

The newly added paragraph at the end of the “Discussion” section is also listed below for your reference.

Overall, maize yield improvement depends on complex interactions among genetics, environment and management. Based on the data-driven approach and field trials, we demonstrate that China would be self-sufficient in maize with current cropping areas in the 2030s through denser planting in combination with selecting best-suited hybrid varieties and soil improvements under future climate. The findings also indicate that high-quality soils with higher SOM could moderate the impact of climate change on OPD and thus improve maize yield. The OPD reflect the density-genetics interaction in this study. In the future, the interactions between hybrid and density could be further enhanced for yield. Our results challenge the view that grain yield have reached an attainable maximum in most global areas and provide a workable example for grain yield improvement through dense planting.

The added part at the end of “Introduction” section is also listed below for your reference.

We explore the potential of yield improvement through dense planting in combination with best-suited hybrid varieties and soil improvements under current and future climates through the integration of machine learning and field trials methodologies. Furthermore, the question of whether China can be self-sufficient in maize under projected future climate change given its existing cropping areas and optimal agricultural management in the 2030s is also addressed.

Reviewers' Comments:

Reviewer #1:

Remarks to the Author:

Recommendation: Accept With No Changes

The authors have satisfactorily responded to all my questions and made the necessary changes to the manuscript.

Reviewer #2:

Remarks to the Author:

I'm generally satisfied with how the authors addressed my first round of comments. Thank you for the great revision. In particular, the detailed response letter. Good job!

Reviewer #3:

Remarks to the Author:

The authors did a very good job addressing reviewers comments. The responses are clear. I have no further concerns.